# The low-density lipoprotein receptor and apolipoprotein E associated with CCHFV particles mediate CCHFV entry into cells

Maureen Ritter[1], Lola Canus[1,7], Anupriya Gautam[1,7], Thomas Vallet[1,7], Li Zhong[1,7], Alexandre Lalande [1], Bertrand Boson [1], Apoorv Gandhi[1], Sergueï Bodoirat[1], Julien Burlaud-Gaillard[2,3], Natalia Freitas[1], Philippe Roingeard [2,3], John N. Barr [4], Vincent Lotteau [5], Vincent Legros[1,6], Cyrille Mathieu [1], François-Loïc Cosset [1,8] ✉ & Solène Denolly [1,8] ✉

The Crimean-Congo hemorrhagic fever virus (CCHFV) is an emerging pathogen of the *Orthonairovirus* genus that can cause severe and often lethal hemorrhagic diseases in humans. CCHFV has a broad tropism and can infect a variety of species and tissues. Here, by using gene silencing, blocking antibodies or soluble receptor fragments, we identify the low-density lipoprotein receptor (LDL-R) as a CCHFV entry factor. The LDL-R facilitates binding of CCHFV particles but does not allow entry of Hazara virus (HAZV), another member of the genus. In addition, we show that apolipoprotein E (apoE), an exchangeable protein that mediates LDL/LDL-R interaction, is incorporated on CCHFV particles, though not on HAZV particles, and enhances their specific infectivity by promoting an LDL-R dependent entry. Finally, we show that molecules that decrease LDL-R from the surface of target cells could inhibit CCHFV infection. Our study highlights that CCHFV takes advantage of a lipoprotein receptor and recruits its natural ligand to promote entry into cells.

The Crimean-Congo hemorrhagic fever virus (CCHFV) is a tick-born zoonotic virus, responsible for severe hemorrhagic fever outbreaks in humans, with a case fatality rate of 10–40%, while being asymptomatic in non-human hosts[1]. CCHFV is endemic in Asia, the Middle East, Eastern Europe, Africa, and more recently, Southern Europe[2,3], which corresponds to the geographic distribution of its vector and/or reservoir, mainly *Hyalomma* ticks[4].

CCHFV is an enveloped virus that belongs to the *Nairoviridae* family of the *Bunyavirales* order. The viral genome consists of three single-stranded RNA segments (L, M, and S) of negative or ambisense polarity. The RNA segments exclusively replicate in the cytosol and encode up to five non-structural proteins and four structural proteins, which are the RNA-dependent RNA polymerase L, the nucleoprotein NP, and two envelope glycoproteins (GP) Gc and Gn. The NP protein binds to genomic RNA to form, together with the viral polymerase, the pseudo-helical ribonucleoproteins (RNPs) inside the virions. Inserted on the viral envelope, the Gn and Gc GPs are responsible for the attachment of viral particles to the surface of host cells and their subsequent penetration into the cytosol (reviewed in[5]).

[1]CIRI – Centre International de Recherche en Infectiologie, Univ. Lyon, Université Claude Bernard Lyon 1, Inserm, U1111, CNRS, UMR5308, ENS de Lyon, F-69007 Lyon, France. [2]Inserm U1259, Morphogénèse et Antigénicité du VIH et des Virus des Hépatites (MAVIVH), Université de Tours and CHRU de Tours, 37032 Tours, France. [3]Université de Tours and CHRU de Tours, Plateforme IBiSA de Microscopie Electronique, Tours, France. [4]Faculty of Biological Sciences and Astbury Centre for Structural Molecular Biology, University of Leeds, Leeds LS2 9JT, UK. [5]Laboratory P4-Jean Mérieux, Inserm, Lyon, France. [6]Campus vétérinaire de Lyon, VetAgro Sup, Université de Lyon, Lyon, Marcy-l'Etoile, France. [7]These authors contributed equally: Lola Canus, Anupriya Gautam, Thomas Vallet, Li Zhong. [8]These authors jointly supervised this work: François-Loïc Cosset, Solène Denolly. ✉e-mail: francois-loic.cosset@ens-lyon.fr; solene.denolly@inserm.fr

The cellular receptors and co-factors involved in CCHFV entry to host cells remain poorly identified. Only the human C-type lectin DC-SIGN and the nuclear factor Nucleolin have been proposed to be involved in CCHFV entry[6,7], but they might not be sufficient for CCHFV entry. Interestingly, a member of the low-density lipoprotein receptor (LDL-R) family, the low-density lipoprotein receptor-related protein 1 (Lrp1) was recently identified as a critical host entry factor for Rift Valley fever virus (RVFV)[8,9] and Oropouche orthobunyavirus (OROV)[10], two members of the *Bunyavirales* order. In addition, other members of the LDL-R family, i.e., the very-low-density lipoprotein receptor (VLDL-R) and the apolipoprotein E receptor 2 (apoER2) were also recently identified as host factors for cell entry of alphaviruses[11], while LDL-R was identified as host entry factor for hepatitis C virus (HCV)[12–14], hepatitis B virus (HBV)[15], Japanese encephalitis virus (JEV)[16]. Finally, several members of the LDL-R family are involved as receptors in vesicular stomatitis virus (VSV) entry[17,18], suggesting that lipid transfer receptors might be used by different viral families.

We therefore sought to investigate in this study if lipid transfer receptor(s) could be used by CCHFV to promote cell entry. We used CCHF transcription and entry-competent virus-like particles (tecVLP)[19] that can be handled in BSL-2 and that fully mimic viral particles[20] as they contain all the structural proteins and a minigenome segment encoding a reporter protein (Fig. 1a). These particles were previously used for neutralization assays[21] or testing of inhibitors[20]. We also confirmed our results with wild-type (WT) virus, which needs to be manipulated in BSL-4.

Here, we show that LDL-R is a cofactor for CCHFV entry, promoting binding of viral particles to cell surface. In addition, we demonstrate that this binding occurs via apolipoprotein E (apoE), a natural ligand of LDL-R that is found to be incorporated on CCHFV particles.

## Results

### The low-density-lipoprotein receptor (LDL-R) is a cofactor of CCHFV infectivity

Lipid transfer receptors may play significant roles during cell entry for different virus families. Here, we chose Huh-7.5 cells as they are fully permissive for CCHFV infection[22] and express several of such receptors, including the low-density lipoprotein receptor (LDL-R), the LDL receptor-related protein 1 (Lrp1), and the scavenger receptor B1 (SR-B1). While Lrp1 and LDL-R are members of the LDL-R family, SR-BI mediates lipid transfer through a different mechanism. To address whether they could be involved in CCHFV entry, we knocked down (KD) several of either lipid transfer receptor (Fig. 1b) by transduction of Huh-7.5 cells with lentiviral vectors encoding specific shRNA. We next transduced these KD cells with serial dilutions of CCHF tecVLPs harboring a nanoluciferase (nanoLuc) reporter gene (tecVLP-NanoLuc, Fig. 1a) and determined transduction levels by luciferase activity measurement. While the modulation of expression of SR-B1 did not change the transduction levels of tecVLPs, we found that down-regulation of LDL-R could significantly reduce transduction of Huh-7.5 cells whereas down-regulation of Lrp1 promoted transduction of Huh-7.5 cells (Fig. 1c). Interestingly down-regulation of Lrp1 did not affect total level of LDL-R (Supplementary Fig. 1a) but induced an increase of LDL-R at cell surface (Supplementary Fig. 1b), which might explain why down-regulation of Lrp1 promoted transduction of tecVLPs.

Next, we aimed at clarifying the role of LDL-R in CCHFV infection. First, we analyzed the effect of blocking of LDL-R present at the surface of Huh-7.5 cells by using an LDL-R antibody added before transduction with serially diluted CCHF tecVLPs harboring a GFP reporter gene (tecVLP-GFP, Fig. 1a) or control particles. As positive control, we used vesicular stomatitis virus glycoprotein pseudoparticles (VSVpp), i.e., lentiviral vector particles pseudotyped with VSV-G, whose transduction depends on LDL-R for entry into cells[17,18], whereas as negative control, we used lentiviral vector particles pseudotyped with the Env

glycoprotein from amphotropic murine leukemia virus (MLVpp), whose entry depends on interaction with the PiT-2, a type III sodium-dependent phosphate transporter[23]. The transduction titers were assessed upon determination of the percentage of positive cells by flow cytometry at 48 h post-transduction (p.t.) (Supplementary Fig. 1c). In agreement with the results of LDL-R KD, we found that LDL-R blocking inhibited transduction of both CCHF tecVLP and VSVGpp particles in an LDL-R antibody dose-dependent manner but did not affect MLVpp transduction (Fig. 1d). Then, we sought to confirm this result using WT CCHFV of the Ibar10200 strain that we produced in Huh-7.5 cells in a BSL4. Upon infection of Huh-7.5 cells with serial dilutions of infectious virus stocks and subsequent assessment of the levels of infection at 24 h post-infection (p.i.), via quantification of viral RNAs in infected cell lysates (Fig. 1a, Supplementary Fig. 1d), we confirmed that the blocking of LDL-R could dose-dependently inhibit CCHFV infection (Fig. 1e).

We also tested the dependency to LDL-R of Hazara virus (HAZV), another member of the genus *Orthonairovirus*, using a GFP-expressing recombinant virus[24] produced in Huh-7.5 cells. Interestingly, the blocking of LDL-R at the surface of Huh-7.5 cells did not impair HAZV infection as assessed by flow cytometry (Fig. 1d), suggesting that LDL-R is not a pan-*Orthonairovirus* entry factor.

LDL-R shares a highly homologous structure with the very low-density lipoprotein receptor (VLDL-R), which is widely expressed with the exception of hepatocytes, including Huh-7.5 cells, under normoxic conditions[25]. We therefore tested the effect of its ectopic expression on CCHF tecVLP transduction using Huh-7.5 cells transduced with a VLDL-R-encoding lentiviral vector. Interestingly, expression of VLDL-R increased the transduction levels of tecVLP-NanoLuc particles by up to 3-fold (Fig. 1f), indicating that alike the LDL-R, VLDL-R may promote CCHFV entry.

Next, we aimed at confirming the LDL-R-dependent CCHFV entry in primary human hepatocytes (PHH), which express LDL-R (Fig. 2a) and could be transduced by tecVLP-NanoLuc particles (Supplementary Fig. 1e). The read-out was performed at 24 h to maximize the level of signal in these primary cells. We found that the transduction of PHH was sensitive to LDL-R blocking (Fig. 2b). Then, we tested the involvement of LDL-R for CCHFV entry in cells from different tissues and species. First, we tested other human cells than Huh-7.5 hepatoma cells, either A549 lung epithelial cells or TE-671 rhabdomyosarcoma cells, which could readily be transduced by tecVLP-NanoLuc particles (Supplementary Fig. 1f). Interestingly, we found that transduction of both A549 and TE-671 cells, which express LDL-R (Fig. 2a), was sensitive to LDL-R blocking (Fig. 2c). Second, as CCHFV can also infect cattle, we tested the LDL-R dependency of CCHFV entry in bovine cells, either EBL embryonic lung cells or MDBK kidney cells, which were found permissive to tecVLP-NanoLuc transduction (Supplementary Fig. 1g). Yet, while the LDL-R blocking antibody could bind LDL-R expressed at the surface of bovine cells (Fig. 2a), it had no effect on tecVLP-NanoLuc transduction in these blocking assays (Fig. 2d), thus suggesting that CCHFV infection in EBL and MDBK cells may not depend on LDL-R.

Altogether, these results suggested that LDL-R is used by CCHFV for infection of human cells but not of bovine cells.

### LDL-R is involved at cell entry steps of CCHFV

Since the assessment of the levels of transduction of CCHF tecVLPs requires both cell entry of viral particles and subsequent transcription and replication of their minigenome, we sought to determine if LDL-R is involved at entry vs. transcription/replication steps. To discriminate either possibility, we added LDL-R antibody at different time points before and/or after transduction of Huh-7.5 cells (Fig. 3a). We found that while the addition of the antibody either before or concomitantly to the transduction step inhibited tecVLP-GFP transduction efficiency to up to 80%, the addition of LDL-R antibody from 2 h post-transduction had no effect on transduction efficiency (Fig. 3b),

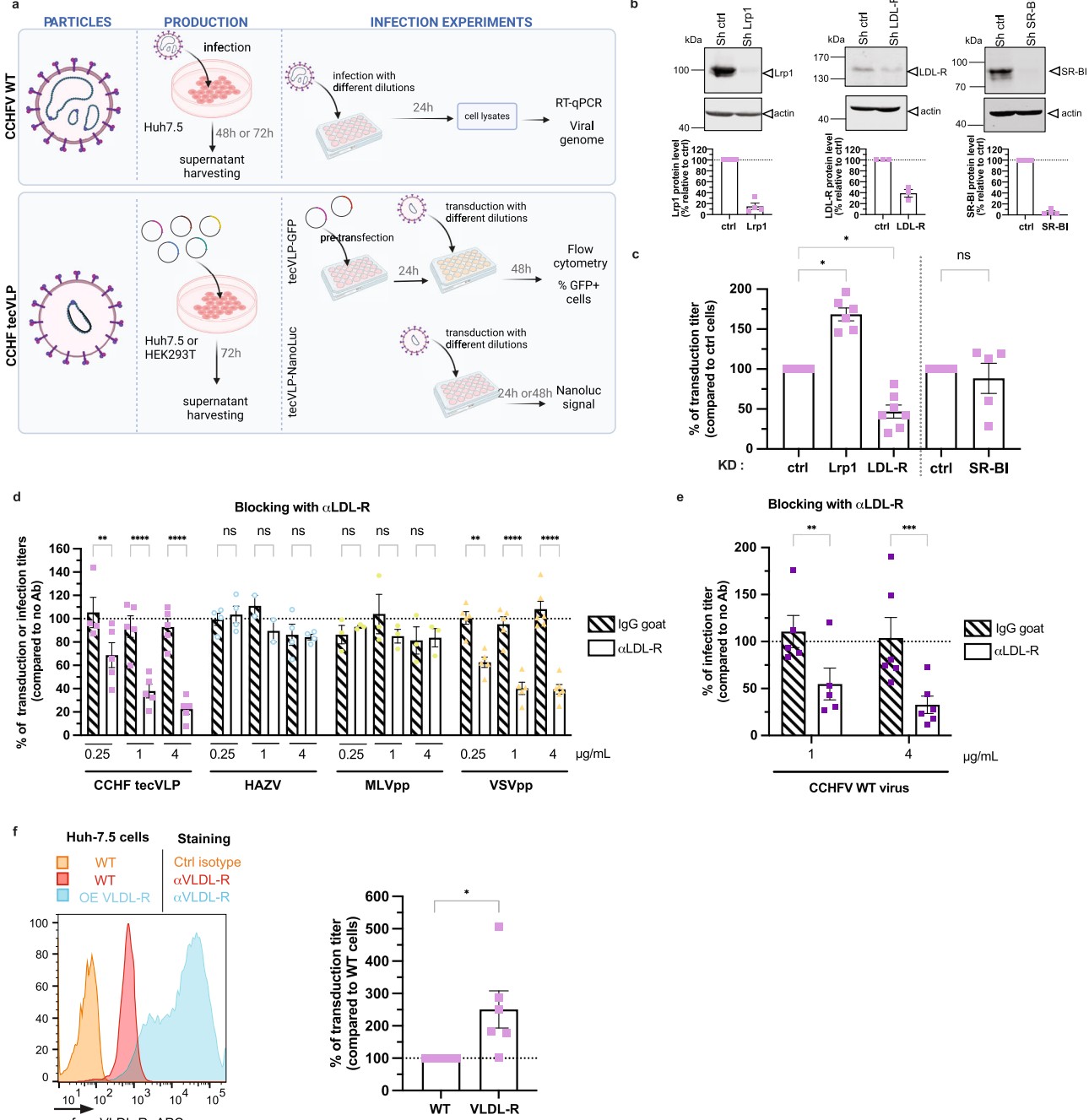

**Fig. 1 | LDL-R is a cofactor of CCHFV infectivity. a** WT CCHFV, manipulated in BSL4, was produced in Huh-7.5 cells, whereas CCHF tecVLPs were produced in Huh-7.5 or HEK-293T cells. Infection or transduction assays were performed with serial dilutions. The level of infection was determined by RNA measurement in infected cells lysates. For tecVLP-GFP, target cells were pre-transfected with CCHFV NP and L expression vectors to amplify the GFP signal by minigenome replication and the level of transduction was assessed by flow cytometry. For tecVLP-NanoLuc, the level of transduction was assessed by measurement of nanoLuc levels. Created with Biorender.com. **b** Western blot analysis of cell lysates from Huh-7.5 cells transduced with lentiviral vectors allowing expression of control shRNA or shRNA targeting Lrp1 or LDL-R or SR-B1 (top). Representative image of 3 experiments. Quantification of the abundance of corresponding receptors (bottom). **c** Cells described in (**b**) were transduced with CCHF tecVLP-NanoLuc. Independent experiments: N = 5 SR-BI; N = 6 Lrp1; N = 7 LDL-R. Kruskal-Wallis test with Dunn's multiple comparisons (ctrl vs. Lrp1: p = 0.0278, ctrl vs. LDL-R: p = 0.0498, ctrl vs. SR-BI: p > 0.9999). **d** Huh-7.5 cells were incubated with different concentration of LDL-R antibody (open bars)

or control isotype (IgG goat, dashed bars) for 1 h at 37 °C before transduction with CCHF tecVLP-GFP (pink), MLVpp (green), and VSVpp (yellow) or infection with HAZV (blue). Independent experiments: N = 3 MLVpp; N = 5 CCHFV tecVLPs and VSVpp; N = 4 HAZV (0.25μg/mL and 4μg/mL) and N = 2 HAZV (1μg/mL). Two-way ANOVA test with Sidak's multiple comparisons (αLDL-R vs. IgG: CCHFV tecVLPs, 0.25μg/mL p = 0.0091, 1μg/mL p < 0.0001, 4μg/mL p < 0.0001; HAZV, 0.25μg/mL p > 0.9999, 1μg/mL p = 0.8951, 4μg/mL p > 0.9999; MLVpp, 0.25μg/mL p > 0.9999, 1μg/mL p = 0.8343, 4μg/mL p > 0.9999; VSVpp, 0.25μg/mL p = 0.0028, 1μg/mL p < 0.0001, 4μg/mL p < 0.0001). **e** Same experiment using WT CCHFV. N = 4 (1μg/mL) or N = 5 (4μg/mL) independent experiments. Two-way ANOVA test with Sidak's multiple comparisons (αLDL-R vs. IgG: 1μg/mL p = 0.0042, 4μg/mL p = 0.0004). **f** Huh-7.5 cells stably expressing FLuc cells were transduced with a lentiviral vector allowing expression of VLDL-R. Surface expression of VLDL-R assessed by flow cytometry (left). Cells were then transduced with CCHF tecVLP-NanoLuc. N = 6 independent experiments. One sample t-test (two-tailed) p = 0.0467. Data are represented as means ± SEM.

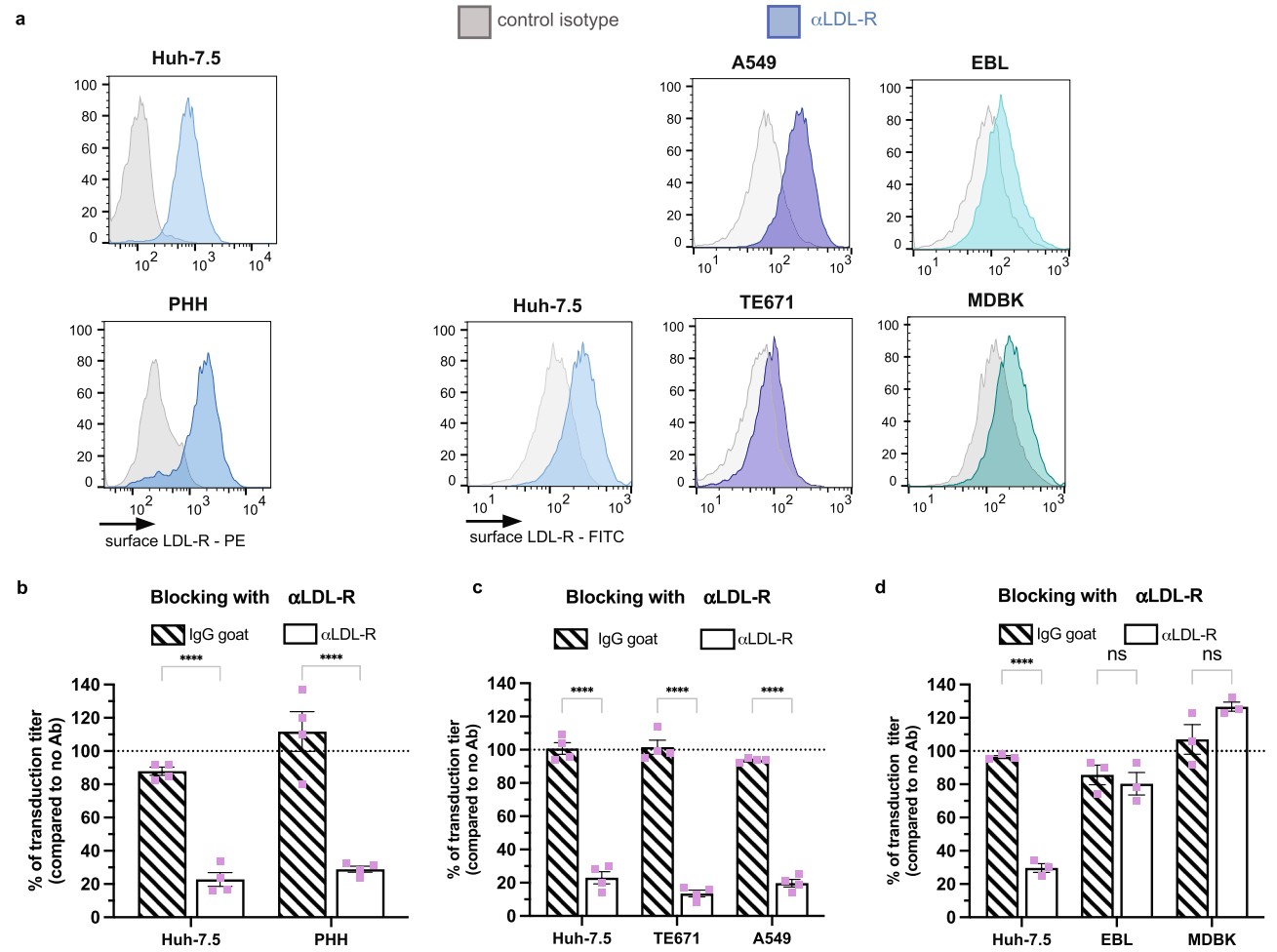

**Fig. 2 | LDL-R entry functions are conserved for infection of different human cells but not for bovine cells. a** Expression of LDL-R at the surface of Huh-7.5, A549, TE-671, EBL, MDBK, and PHH cells assessed by flow cytometry. **b** Huh-7.5 cells or PHH were incubated with 4 μg/mL of LDL-R antibody (open bars) or IgG goat (dashed bars) for 1 h at 37 °C before transduction with CCHF tecVLP-NanoLuc. $N = 4$ independent experiments. Two-way ANOVA test with Sidak's multiple comparisons (αLDL-R vs. IgG: Huh-7.5 $p < 0.0001$; PHH $p < 0.0001$). **c** Same as (**b**) with Huh-7.5,

TE-671, A549 cells with harvesting at 48 h post-transduction (p.t.). $N = 4$ independent experiments. Two-way ANOVA test with Sidak's multiple comparisons (αLDL-R vs. IgG: Huh-7.5 $p < 0.0001$; TE671 p $< 0.0001$; A549 $p < 0.0001$). **d** Same as (**c**) with Huh-7.5, EBL, MDBK cells. $N = 3$ independent experiments. Two-way ANOVA test with Sidak's multiple comparisons (αLDL-R vs. IgG: Huh-7.5 $p < 0.0001$; EBL $p = 0.8740$; MDBK $p = 0.0715$). Data are represented as means ± SEM.

hence suggesting that LDL-R is involved at an entry step rather than at a later step of transcription/replication.

We thus hypothesized that LDL-R could serve as a CCHFV entry factor through its expression at the cell surface. To test this hypothesis, we incubated tecVLP-GFP or WT CCHFV particles with a soluble recombinant form of LDL-R (sLDL-R) before transduction or infection of Huh-7.5 cells. We used VSV pseudoparticles (VSVpp) as a positive control and amphotropic murine leukemia virus (MLV) pseudoparticles (MLVpp) as a negative control. While a soluble form of CD81 (CD81-LEL)[26] used as control had no effect on transduction, we found that, like for VSVpp, sLDL-R inhibited CCHFV infection in a dose-dependent manner in both tecVLP-GFP transduction (Fig. 3c) and WT CCHFV infection (Fig. 3d) assays, hence suggesting that sLDL-R could prevent cell entry through interaction with viral particles. Like for the LDL-R blocking experiment (Fig. 1d), we did not observe any impact of sLDL-R neutralization on HAZV infection (Fig. 3c). Note that while the blocking of LDL-R with an antibody impaired VSVpp and CCHF tecVLP at similar levels (Fig. 1d), sLDL-R impaired CCHFV entry at a lesser extent as compared to VSVpp (Fig. 3c, d). This difference between either virus could be due to a different role or affinity of LDL-R for the two types of viral particles. Alternatively, this could also be due to the production of CCHF tecVLPs in Huh-7.5 cells that express competitors

for binding to sLDL-R, such as apoB or apoE, which is not the case for HEK-293T cells that were used to produce VSVpp.

The above data suggested that CCHFV could bind to LDL-R. To test this hypothesis, we incubated CCHF tecVLPs with sLDL-R or CD81-LEL before the capture of these soluble receptors with Ni-NTA beads, as both soluble proteins harbor a 6xHis tag, and determination of the levels of co-captured viral genomes by qPCR. Interestingly, we found that we could capture about 7-fold more CCHF tecVLP RNAs with sLDL-R than with CD81-LEL (Fig. 3e, left). We repeated the same experiment with WT CCHFV and HAZV. While HAZV could not be captured by either protein, we confirmed that WT CCHFV RNAs could be co-captured with sLDL-R (Fig. 3e, right).

Altogether, these results indicated that LDL-R promotes CCHFV entry through the binding of viral particles.

**The exchangeable apolipoprotein E mediates CCHFV entry**
Next, we aimed to confirm the role of LDL-R in CCHFV entry using VSV pseudotyped with CCHFV glycoproteins[27]. Interestingly, while the blocking of LDL-R with LDL-R antibody had a strong effect on the transduction of control VSV particles pseudotyped with VSV-G with over 80% of inhibition, its inhibitory effect on VSV particles harboring CCHFV GPs, of up to 20%, was not only milder than for the former

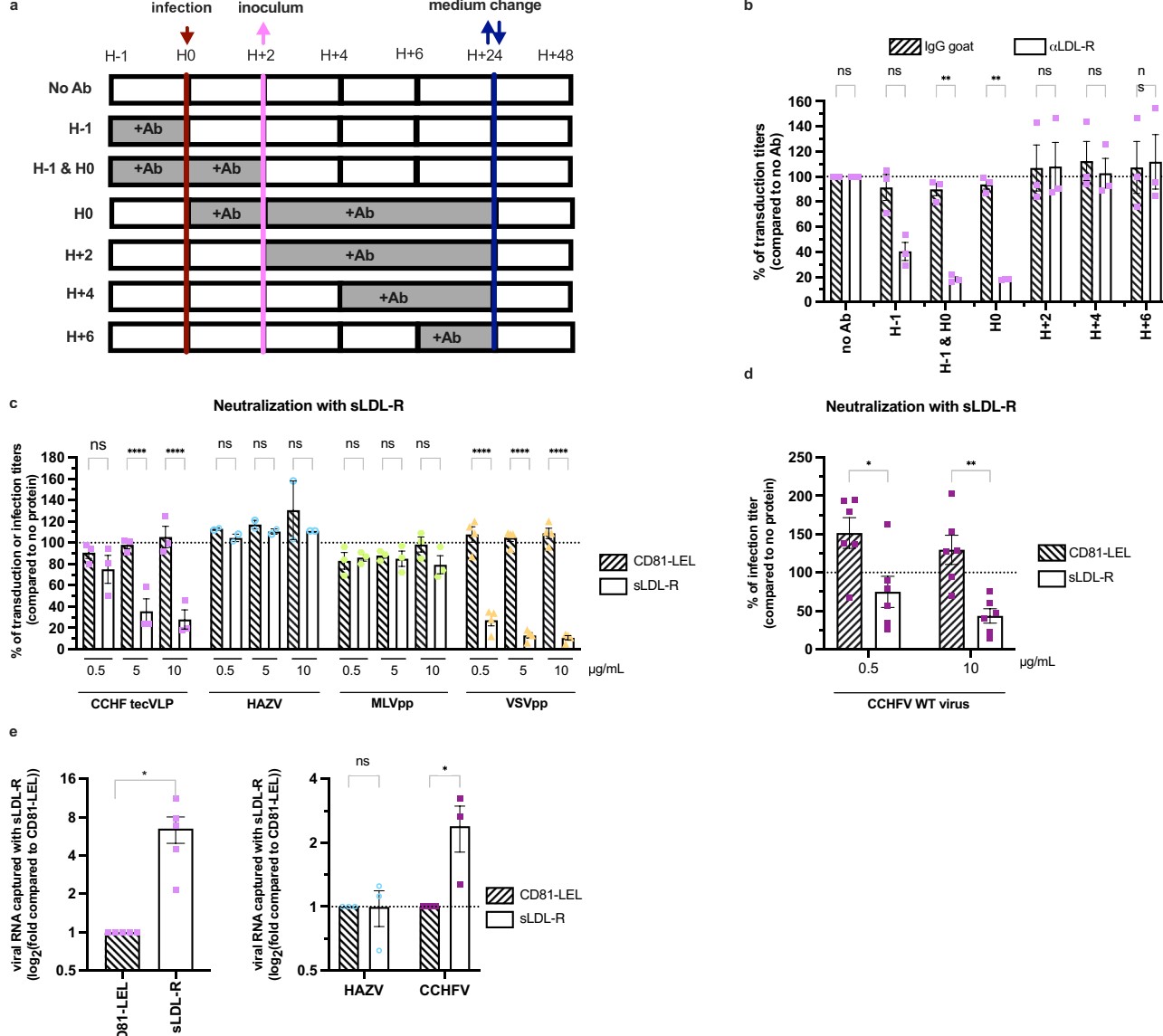

**Fig. 3 | LDL-R promotes CCHFV entry. a** Huh-7.5 cells were incubated with 4 μg/mL of LDL-R antibody or control isotype before, during or after transduction with CCHF tecVLPs GFP as indicated in grey. **b** Percentage of transduction titers of CCHF tecVLP-GFP relative to control isotype as in the experimental set up described in (**a**). *N* = 3 independent experiments. Two-way ANOVA test with Dunnett's multiple comparisons (αLDL-R vs. IgG: no Ab *p* > 0.9999; H-1 *p* = 0.0505; H-1 and H0 *p* = 0.0026; H0 *p* = 0.014; H + 2 *p* > 0.9999; H + 4 *p* = 0.9980; H + 6 *p* > 0.999). **c** CCHF tecVLP-GFP (pink), MLVpp (green), VSVpp (yellow) or HAZV (blue, *N* = 2) were incubated for 1 h at room temperature with soluble LDL-R (sLDL-R, open bars) or with soluble CD81 (CD81-LEL, dashed bars) at different concentrations before transduction or infection of Huh-7.5 cells. Independent experiments: *N* = 3 CCHF tecVLP and MLVpp; N = 4 VSVpp; *N* = 2 HAZV. Two-way ANOVA test with Sidak's multiple comparisons (sLDL-R vs. CD81-LEL: CCHF tecVLPs, 0.5μg/mL *p* = 0.8354, 5μg/mL *p* < 0.0001, 10μg/mL *p* < 0.0001; HAZV, 0.5μg/mL *p* = 0.9998, 5μg/mL

*p* > 0.999, 10μg/mL *p* = 0.8206; MLVpp, 0.5μg/mL *p* > 0.9999, 5μg/mL *p* > 0.9999, 10μg/mL *p* = 0.6061; VSVpp, 0.5μg/mL *p* < 0.0001, 5μg/mL *p* < 0.0001, 10μg/mL *p* < 0.0001). **d** Same experiment using WT CCHFV. Media was removed 1 h post-infection (p.i.) and cells were lysed 24 h p.i. The level of infectivity was quantified by RT-qPCR. *N* = 6 independent experiments. Two-way ANOVA test with Sidak's multiple comparisons (sLDL-R vs. CD81-LEL: 0.5μg/mL *p* = 0.0125, 10μg/mL *p* = 0.0053). **e** CCHF tecVLPs (left) or CCHFV or HAZV (right) were incubated with sLDL-R or CD81-LEL (both expressing a 6xHis tag) for 1 h at RT before capture using magnetic beads. The levels of viral RNA co-captured were determined by RT-qPCR. One sample t-test (two-tailed) for CCHF tecVLPs (*N* = 5 independent experiments, *p* = 0.0227), two-way ANOVA test with Sidak's multiple comparisons for HAZV and CCHFV (*N* = 3 independent experiments, HAZV p > 0.9999; CCHFV *p* = 0.025). Data are represented as means ± SEM.

pseudotypes (Fig. 4a) but was also much lower than for CCHF tecVLPs (Fig. 1d). Since the VSV particles are produced in HEK-293T cells rather than in Huh-7.5 cells like for the CCHF tecVLPs, we wondered if the producer cell type could have an impact on either VSV or tecVLP particles and their dependency to LDL-R. We therefore produced CCHF tecVLPs in HEK-293T cells, which are fully able to produce tecVLPs (Supplementary Fig. 2a, b) albeit at a lower titer than Huh-7.5 cells. Interestingly, the blocking of LDL-R in Huh-7.5 target cells with an

LDL-R antibody had less impact on tecVLPs produced in HEK-293T cells than for tecVLPs produced from Huh-7.5 cells (Fig. 4b, left). To exclude a potential effect on cell compatibility between producer and target cells, we blocked LDL-R, which is expressed in HEK-293T cells (Supplementary Fig. 2c). Using these latter cells as targets, we found that LDL-R blocking only inhibited transduction of tecVLPs produced in Huh-7.5 cells but not those produced in HEK-293T cells (Fig. 4b, right), despite similar transduction levels (Supplementary Fig. 2d, e).

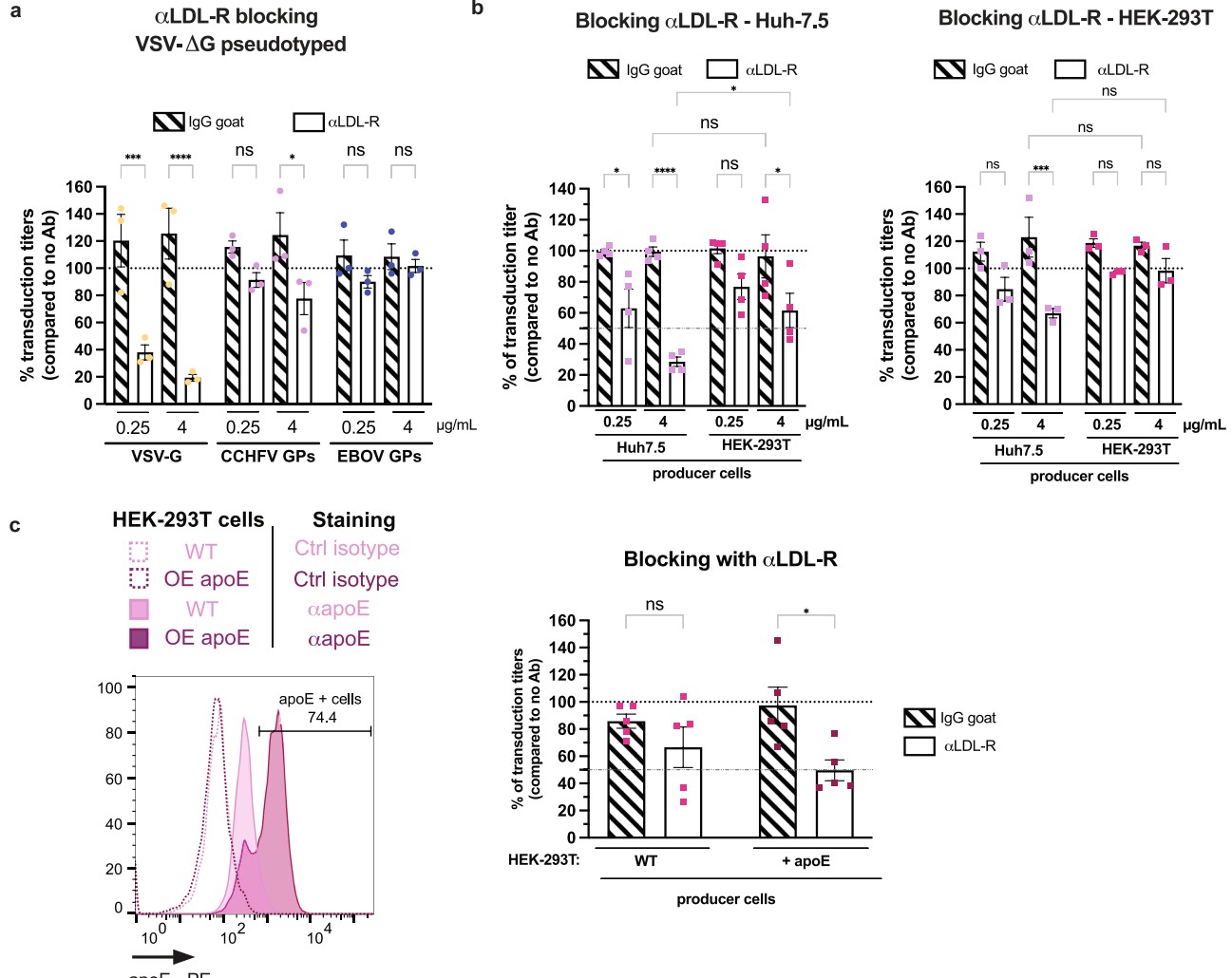

**Fig. 4 | LDL-R dependency of CCHFV entry is influenced by the virus producer cell type. a** Huh-7.5 cells were incubated with 0.25µg/mL or 4µg/mL of LDL-R antibody (open bars) or IgG goat (dashed bars) for 1 h at 37 °C before transduction with VSV-ΔG/GFP particles pseudotyped with VSV-G (yellow), CCHFV GPs (pink) or EBOV GP (blue). N = 3 independent experiments. Two-way ANOVA test with Sidak's multiple comparisons (αLDL-R vs. IgG: VSV-G, 0.25µg/mL p = 0.0001, 4µg/mL p < 0.0001; CCHFV GPs, 0.25µg/mL p = 0.5799, 4µg/mL p = 0.0393; EBOV, 0.25µg/mL p = 0.7924, 4µg/mL p > 0.9988). **b** (left) Same as (**a**) with CCHF tecVLP-GFP particles produced in Huh-7.5 (pink) or HEK-293T (fuchsia) cells. N = 4 independent experiments. Two-way ANOVA test with Sidak's multiple comparisons (αLDL-R vs. IgG: Huh-7.5, 0.25µg/mL p = 0.0106, 4µg/mL p < 0.0001; HEK-293T, 0.25µg/mL p = 0.1528, 4µg/mL p = 0.0159; Huh-7.5 vs. HEK-293T, αLDL-R 4µg/mL p = 0.0245, IgG 4µg/mL p = 0.9999). (right) HEK-293T cells were incubated with 0.25 or 4µg/mL of αLDL-R antibody (open bars) or IgG goat (dashed bars) for 1 h at 37 °C before transduction with CCHF tecVLP-NanoLuc particles produced in Huh7.5 (pink) or HEK-293T (fuchsia) cells. N = 3 independent experiments. Two-way ANOVA test with Sidak's multiple comparisons (αLDL-R vs. IgG: Huh-7.5, 0.25µg/mL p = 0.1085, 4µg/mL p = 0.0004, HEK-293T; 0.25µg/mL p = 0.3004, 4µg/mL p = 0.4791; Huh-7.5 vs. HEK-293T, αLDL-R 4µg/mL p = 0.0545, IgG 4µg/mL p = 0.9999). **c** Intracellular levels of apoE in HEK-293T vs. HEK-239T stably expressing apoE as assessed by flow cytometry (left). CCHF tecVLPs produced in these cells were used for the experiment described as in (**b**) with 4µg/mL of αLDL-R antibody (open bars) or IgG goat (dashed bars) (right). N = 5 independent experiments. Two-way ANOVA test with Sidak's multiple comparisons (HEK-293T: αLDL-R vs. IgG: 0.4196, HEK-293T + apoE: αLDL-R vs. IgG: 0.0152). Data are represented as means ± SEM.

These results therefore indicated that the LDL-R dependency of CCHFV entry could be influenced by the producer cell type. As HEK-293T cells do not express apoE, a natural ligand of LDL-R, in contrast to Huh7.5 cells (Supplementary Fig. 2f), we wondered if apoE might be responsible for the dependency of CCHFV entry to LDL-R. To test this hypothesis, we produced tecVLPs in HEK-293T cells transduced with an apoE lentiviral vector (Fig. 4c, left). Interestingly, ectopic apoE expression in HEK-293T cells increased the titers of tecVLPs (Supplementary Fig. 2g) and the effect of LDL-R blocking upon tecVLP transduction of Huh-7.5 target cells (Fig. 4c, right).

Based on these results, we wondered if and how apoE could influence the entry of CCHFV. First, we tested if apoE antibodies could neutralize CCHF tecVLP transduction or WT CCHFV infection. As a positive control, we used HCV particles, as they can be neutralized by apoE antibodies[28], whereas we used VSVpp and MLVpp as negative controls, since VSV-G is the direct ligand of VSV for LDL-R binding[29] and since MLV Env binds an irrelevant receptor[30]. We incubated either viral particle with apoE antibodies for 1 h before transduction or infection assays. Interestingly, we found a dose-dependent inhibition for both CCHF tecVLP-GFP (Fig. 5a) and WT CCHFV (Fig. 5b) particles by apoE antibodies, reaching up to 80% inhibition in a manner similar to HCV particles, whereas apoE antibodies did not inhibit VSVpp or MLVpp transduction (Fig. 5a). The difference of level of neutralization between CCHF tecVLPs and WT CCHFV could be due a difference of the number of viral particles. Conversely, when we tested the apoE dependency of HAZV infection, we found that neutralization by apoE antibodies did not influence HAZV infection (Fig. 5a).

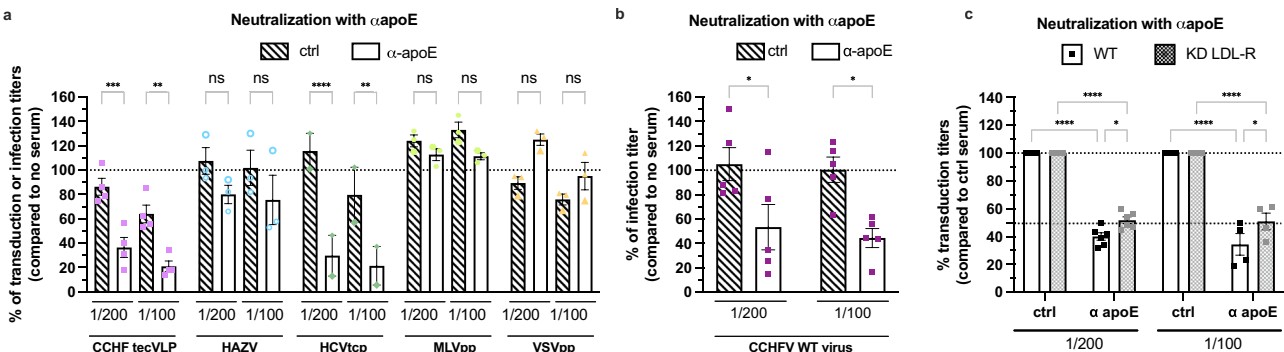

**Fig. 5 | ApoE promotes entry of CCHFV particles. a** CCHF tecVLPs (pink), MLVpp (light green), VSVGpp (yellow), HAZV (blue) or HCVtcp (green) were incubated for 1 h at room temperature with anti-apoE serum or control serum at different dilution before transduction or infection of Huh-7.5 cells. Independent experiments: $N = 4$ CCHF tecVLP; $N = 3$ MLVpp, VSVGpp, and HAZV; $N = 2$ HCVtcp. Two-way ANOVA test with Sidak's multiple comparisons (αapoE vs. ctrl serum: CCHF tecVLP, 1/200 p = 0.0002, 1/100 $p = 0.0018$; HAZV, 1/200 p = 0.2582, 1/100 $p = 0.3151$; HCVtcp, 1/200 $p < 0.0001$, 1/100 $p = 0.0032$; MLVpp, 1/200 $p = 0.9886$, 1/100 $p = 0.5832$; VSVpp, 1/200 $p = 0.0533$, 1/100 $p = 0.7327$). **b** Same experiment as (**a**) using WT CCHFV particles. $N = 5$ independent experiments. Two-way ANOVA test with Sidak's multiple comparisons (ctrl vs. αapoE: 1/100 p = 0.0273, 1/200 $p = 0.0619$). **c** CCHF tecVLP-NanoLuc particles were incubated for 1 h at room temperature with an apoE serum at different dilution before transduction of Huh-7.5 cells stably expressing FLuc and transduced with lentiviral vectors allowing expression of control shRNA or shRNA targeting LDL-R as described in Fig. 1. $N = 6$ independent experiments. Two-way ANOVA test with Sidak's multiple comparisons (WT vs. KD LDL-R: 1/200 p = 0.012, 1/100 $p = 0.0383$; ctrl vs. αapoE: WT 1/200 $p < 0.0001$, WT 1/100 $p < 0.0001$, KD LDL-R 1/200 $p < 0.0001$, KD LDL-R 1/100 $p < 0.0001$). Data are represented as means ± SEM.

These results suggested that apoE plays a crucial role in CCHFV infectivity. Thus, to corroborate the role of apoE in LDL-R-mediated entry, we repeated the apoE neutralization assay using Huh-7.5 target cells in which LDL-R was down-regulated. We found that under such conditions, tecVLP transduction was slightly less efficiently inhibited by apoE antibodies (Fig. 5c), hence suggesting a synergic role of apoE and LDL-R in CCHFV infection.

### ApoE is associated with CCHFV particles and promotes their assembly/secretion and specific infectivity

ApoE is present at the surface of lipoproteins such as LDLs and VLDLs but can also exist in a lipid-free form in the extracellular environnement[31]. Importantly, apoE belongs to the family of exchangeable apolipoproteins, implying that it can be transferred from a lipoprotein to another lipoprotein or to a viral particle, as described for HCV[32,33]. We therefore sought to determine if CCHF tecVLPs harbor apoE at their surface, which would promote entry of CCHFV.

First, we determined if we could capture viral particles with an apoE antibody, as previously shown for HCV[34]. After immunoprecipitation of CCHF tecVLPs or HAZV particles with an apoE antibody, we quantified the captured particles by detecting viral RNA by RT-qPCR. Interestingly, in contrast to HAZV, we found a 16-fold enrichment of CCHF tecVLP RNAs with apoE antibodies relative to control IgGs (Fig. 6a). In addition, we could also detect some CCHF tecVLPs in electron microscopy with immunogold labeling with anti-Gn or anti apoE (Fig. 6b). Moreover, we confirmed the association of WT CCHFV particles with apoE, since we could co-capture both viral RNA and CCHFV Gn and Gc proteins with apoE antibodies (Fig. 6c, d), suggesting an association of apoE to particles containing CCHFV glycoproteins and viral genome.

Second, we produced CCHF tecVLPs in Huh-7.5 cells transduced with a shRNA targeting apoE, which induced a robust loss of apoE expression (Fig. 6e). While apoE KD did not impair the level of expression of CCHFV NP in producer cells (Fig. 6f, top), it resulted in a strong loss of transduction efficiency of CCHF tecVLPs, with a 2-log titer decrease (Fig. 6g, top). To determine if this loss resulted from a defect in assembly *vs.* specific transduction efficiency of particles, we determined the levels of viral RNA in the supernatant. We found that apoE KD impaired by ca. 1-log the secretion of the viral genome (Fig. 6h, top), indicating that apoE plays a role in both assembly/

secretion and specific transduction efficiency of CCHF tecVLP particles. Interestingly, apoE KD had no effect on HAZV production and infectivity (Figs. 6f–h, bottom), which agreed with the results of lack of LDL-R and apoE dependency of this virus (Figs. 1c, 3c and e, 5a).

Altogether, these results indicated that CCHFV particles could incorporate apoE, which may therefore provide a ligand of LDL-R at the surface of the viral particles, and that apoE is a pro-viral factor for assembly/secretion and specific infectivity of CCHFV particles.

### Molecules impairing LDL-R surface levels prevent CCHFV infection

Finally, we tested if molecules that regulate LDL-R surface levels could modulate CCHFV entry, aiming at proposing possible ways to prevent CCHFV infection. Using the proprotein convertase subtilisin-like kexin type 9 (PCSK9) that inhibits LDL-R recycling[35] and therefore decreases LDL-R exposition at the cell surface (Fig. 7a, b), without altering cell viability (Fig. 7c), we found that pre-treatment of cells with PCSK9 impaired transduction of both VSVpp and CCHF tecVLPs without affecting MLVpp transduction and HAZV infection (Fig. 7d).

We also tested Berbamine - bis-benzylisoquinoline (BBM), an alkaloid isolated from the plant *Berberis amurensis* (used in traditional Chinese medicine), that was reported to inhibit JEV by altering cell surface LDL-R level[16]. Again, we showed that pre-treatment of the cells with BBM decreased LDL-R levels at the cell surface (Fig. 7e, f) without altering cell viability (Fig. 7g) and impaired transduction of CCHF tecVLPs and VSVpp, without affecting transduction of MLVpp (Fig. 7h). Of note, BBM also impaired to some extent HAZV infection (Fig. 7h), which might be due to a broad effect of BBM on cellular pathways.

These results highlighted that molecules modulating LDL-R surface levels could be used to prevent CCHFV infection.

Altogether, our study identified LDL-R as a factor promoting CCHFV infection via binding of the viral particles. We also showed that CCHFV particles incorporate a natural ligand of LDL-R, apoE, and that this factor promotes the LDL-R-dependent entry (Fig. 8), as well as assembly of viral particles.

### Discussion

Our results highlight the role of LDL-R and apoE as entry factors of CCHFV. Importantly, our findings using the CCHF tecVLP assay, which relies on VLPs that contain all the structural proteins but only a mini-genome segment encoding a reporter protein[19,20,22], were fully

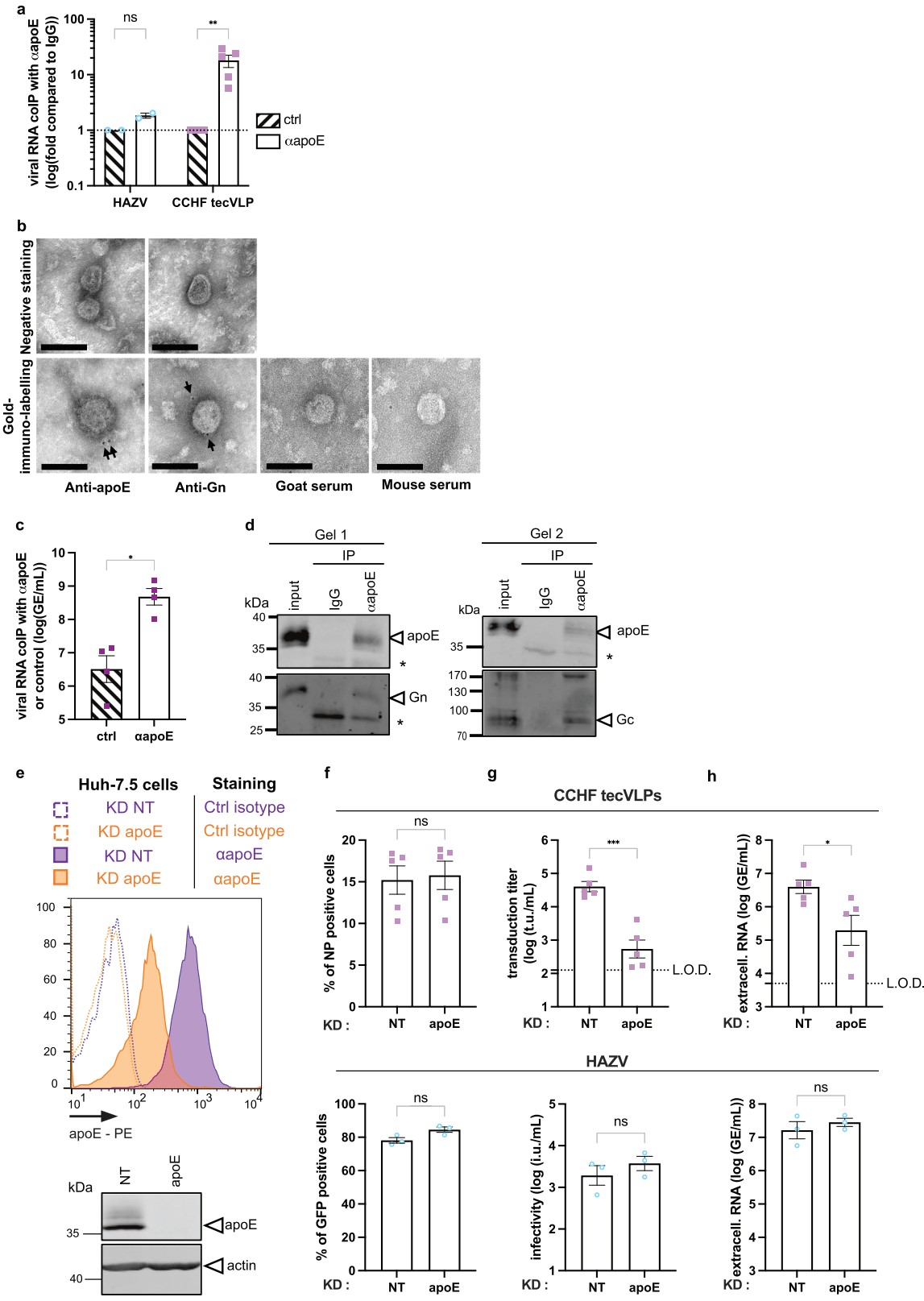

confirmed with WT CCHFV handled in BSL4. This indicates that although they do not allow full investigation of CCHFV properties, the former particles provide a bona fide assay to investigate cell entry pathways and receptors of CCHFV. In agreement with these findings, a recent study also identified LDL-R as an entry factor of CCHFV[36] (published while this article was in revision) using a different CCHFV strain than ours. The LDL-R is the prototype member of the 'LDL-R family', which regroups structurally related endocytic receptors that mediate lipid transfer to cells. The ectodomains of the members of this family share high sequence similarity and capacity to bind to a large variety of ligands[37]. They are composed of cysteine-rich repeats, which are repeats of their ligand-binding domains, and of EGF-like modules and β-propellers, which are required for the pH-dependent release of their ligands following internalization.

**Fig. 6 | ApoE is associated with CCHFV particles and contribute to assembly/secretion and specific infectivity. a** Level of CCHFV minigenome RNA or HAZV RNA co-immunoprecipitated with an apoE serum *vs.* control IgGs and quantified by RT-qPCR. Results from $N = 2$ (HAZV) or $N = 5$ (CCHF tecVLP) independent experiments are presented as fold enrichment with apoE antibodies relative to control IgGs. Two-way ANOVA test with Sidak's multiple comparisons (HAZV $p = 0.9895$, CCHF tecVLPs $p = 0.0036$). **b** Representative electron microscopy images of tecVLPs with simple negative stain (top) or with immunogold labelling with anti-Gn or anti-apoE antibodies or control antibodies (bottom). Scale bar represents 100 nm. Representative images from 2 experiments. **c** CCHFV particles were immunoprecipitated with an apoE serum vs. control IgGs. $N = 4$ independent experiments. Mann-Whitney test (two-tailed, $p = 0.0286$). **d** CCHFV particles were immunoprecipitated with an apoE serum *vs.* control IgGs and analyzed by western blot for apoE and Gn or Gc detection. Asterisks indicated unspecific bands from antibodies light chains. Representative image of 4 independent experiments.

**e** Intracellular levels of apoE as assessed by flow cytometry and Western blot of cells transduced with shRNA control (NT) or targeting apoE. Representative image of 3 independent experiments. **f** Cells described in (**e**) were used for the production of CCHF tecVLPs or HAZV particles as described in Methods. Percentage of CCHFV NP transfected cells (top) or HAZV-eGFP expressing cells (bottom). Unpaired t-test (two-tailed, $p = 0.8219$) for CCHFV and Mann-Whitney test (two-tailed, p = 0.1) for HAZV. **g** Transduction efficiency of CCHF tecVLPs (top) or infectivity of HAZV (bottom) particles produced in cells described in (**e**) as assessed by flow cytometry. Unpaired t-test (two-tailed, p = 0.0003) for CCHFV and Mann-Whitney test (two-tailed, $p = 0.4$) for HAZV. **h** Level of secreted viral RNA of tecVLPs (top) or HAZV (bottom) assessed by RT-qPCR. Unpaired t-test (two-tailed, $p = 0.0296$) for CCHFV and Mann-Whitney test (two-tailed, $p = 0.7$) for HAZV. For (**f–h**), $N = 5$ independent experiments for CCHFV tecVLPs and $N = 3$ for HAZV. Data are represented as means ± SEM.

Interestingly, entry of unrelated bunyaviruses, including RVFV and OROV was recently shown to involve Lrp1, a member of the LDL-R family[8–10], which does not appear to be a proviral entry factor for CCHFV (Fig. 1c). Yet, together with our results that LDL-R acts as a cofactor for CCHFV entry though not for HAZV, these findings imply that different binding and/or post-binding functions of members of the LDL-R family have been coopted by bunyaviruses in a virus-specific manner to promote their entry into cells. While LDL-R is mainly involved in the endocytosis of triglyceride- and cholesterol-containing lipoprotein particles, Lrp1 mediates the endocytosis of different types of ligands especially in the liver[25]. Overall, the members of the LDL-R family can bind different types of proteins or factors, suggesting that these receptors could, at least, act as capture molecules. Indeed, as above-mentioned, OROV and RVFV particles were shown to bind Lrp1 whereas we found that CCHFV particles can bind LDL-R. Furthermore, a recent study suggested that Lrp1 plays a role in RVFV endocytosis although it was unclear if this occurs via direct or indirect interactions with viral particles[8].

Importantly, the usage of LDL-R family members as cell entry cofactors is not restricted to bunyaviruses since several other viruses seem to hijack members of this family, such as HCV for VLDL-R[12,38] and LDL-R[13,14,28], HBV for LDL-R[15], alphaviruses for VLDL-R and apoER2[11], VSV for LDL-R[17,18], Dengue virus and JEV for LDL-R[16,39], as well as some rhinoviruses for LDL-R and VLDL-R[40,41].

Altogether, these previous studies combined with our report underscore a wide-ranging role for receptors of the LDL-R family in virus entry. Yet, how these factors promote virus entry remains poorly defined. The current evidence suggests that overall, most of these factors may not act as bona fide viral receptors but rather, as above-discussed, as crucial co-factors of virus entry by promoting the capture of the viral particles at the cell surface or alternatively, their endocytosis. On the other hand, for some of these viruses, it is not even clear if the viral particles bind to these cofactors. While some viruses such as RVFV[9], OROV[10] or VSV[29] seem to directly bind these receptors via their glycoproteins, some other viruses hijack cellular proteins like apoE as ligand cofactor for binding LDL-R, as shown in this study for CCHFV and as previously shown for HCV[28] and HBV[42]. We may therefore speculate that viruses that can replicate in hepatocytes could have evolved to easily hijack some lipoprotein components, such as apoE or alternative exchangeable apolipoproteins[43,44] that are produced in the same cells, either during their assembly or secretion or from the extracellular environment (see below). In contrast, other viruses could have taken advantage of the capacity of LDL-R family members to bind to a large variety of ligands via a relatively unspecific mechanism. Indeed, for some of these ligands, the interactions can involve electrostatic interactions between conserved acidic residues or tryptophans on LDL-R repeats with basic residues on the ligands (reviewed in ref. 37), as shown for human rhinovirus serotype 2 (HRV2) and VLDL-R[41].

Especially, while OROV and RVFV Gn GP may directly bind Lrp1[9,10], our results indicate that apoE, a natural ligand of members of the LDL-R family, is incorporated onto CCHFV particles and promotes LDL-R dependent entry (Fig. 8). On the other hand, the CCHFV Gc GP may also directly bind LDL-R[36], suggesting different though not mutually exclusive mechanisms or, alternatively, a tripartite molecular interaction developed by CCHFV to interact with LDL-R and promote cell entry. This may depend on the presence of apoE as well as on other host factors that may be expressed, or not, in virus-producer cells and that could possibly directly interact with CCHFV GPs or virion surface. Association of CCHFV particles with apoE is reminiscent of the properties of HCV and HBV[28,34,42,45]. Indeed, in the case of HCV, previous studies indicated that apoE association with the viral particles allow them to bind different entry factors such as heparan sulphate proteoglycans (HSPG)[46,47], which act as capture molecules[48], but also to lipid transfer receptors such as LDL-R[28] and SR-BI[34,45]. Furthermore, binding of HCV particles to SR-BI was shown to be mediated by either ApoE or HCV E2 surface glycoprotein[45], which is reminiscent of the situation with CCHFV particles that may interact with the LDL-R in a Gc-dependent manner[36] or, alternatively, in an apoE-dependent manner (this report). Finally, our results also underscore that species-specific determinants could be important in the above molecular interactions and their outcome in virus infectivity. For example, previous results indicated that apoE expressed in Vero cells does not allow the production of infectious HCV particles[49], which could be due to a lack of interaction of this simian apoE with HCV E2 or alternatively with human LDL-R. This suggests possibilities to explain why the infectivity of CCHFV grown in Vero cells seemed to rely on the sole interaction between Gc and LDL-R[36] whereas the infectivity of CCHFV grown in human cells, such as Huh-7.5 cells, depends on or is strengthened by human apoE (this report).

As apoE is a high affinity ligand for most receptors of the LDL-R family[50], whether it also acts as a ligand recruited by viral particles of the above-mentioned viruses that use lipid transfer receptors remains an open question. In this respect, it is surprising that only LDL-R though not Lrp1 and SR-B1 acts as an entry factor of CCHFV. While further studies are needed to understand these differences, one possibility could be that Gc binds specifically to LDL-R rather than the other family members, and that apoE would stabilize this binding and promote entry through a bi-partite apoE/Gc interaction allowing productive LDL-R dependent entry. Another possibility could be that LDL-R may participate to the formation of a receptor complex through a specific association with alternative putative Gn/Gc receptor(s) (Fig. 8). On the other hand, as the location of the viral binding site on the receptor is a critical determinant of membrane fusion[51], one could speculate that should apoE allow binding of CCHFV particles to Lrp1 and SR-B1, it may not provide the optimal distance between viral and target cell membranes, in agreement with a much longer extracellular domain for Lrp1 than for LDL-R[52]. Finally, as shown by the results with

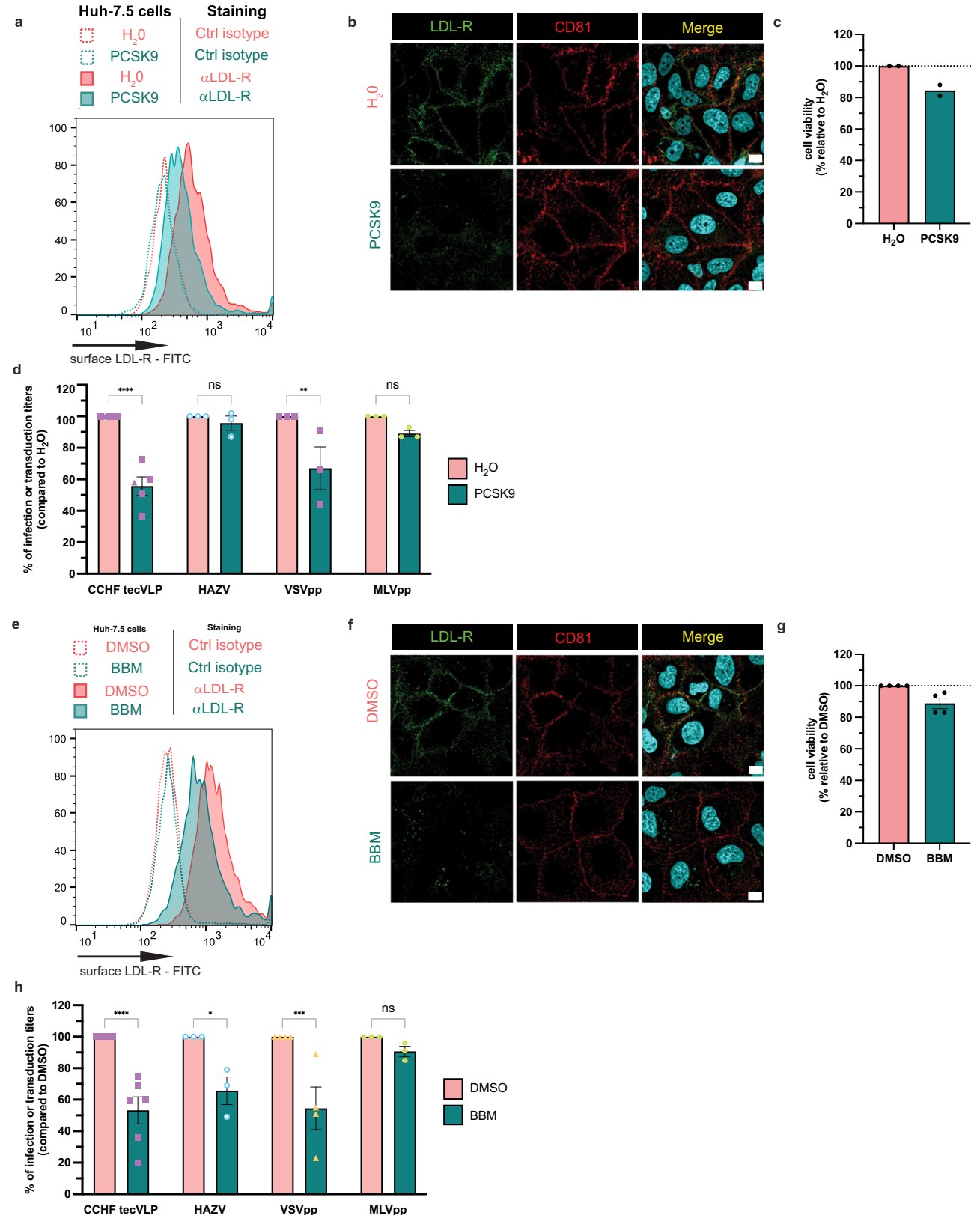

HCV particles, whose endocytosis by LDL-R[13] or increasing endocytosis by SR-BI[53] leads to non-productive or reduced entry into cells, that Lrp1 can be endocytosed more actively than LDL-R[54] also raises the possibility that, even if CCHFV binds to Lrp1, this could lead to a non-productive entry. This is supported by our data suggesting that Lrp1 seems to be antiviral for CCHFV infection (Fig. 1c), though it could

also be due to a regulation of the level of LDL-R at the cell surface (Supplementary Fig. 1b).

How apoE is recruited on CCHFV particles remains ill-defined. Our results suggest that its incorporation on viral particles may occur during their production from hepatocytes in which apoE is expressed. This could involve apoE interactions with CCHFV determinants such as

**Fig. 7 | Molecules impairing LDL-R surface level impaired CCHFV infection.**
**a** Cell surface expression of LDL-R of cells treated with 0 vs. 10 μg/mL of PCSK9 for 3 h at 37 °C. Control isotypes are depicted in dotted lines. Representative image of $N = 3$ independent experiments. **b** Cell surface expression of LDL-R or CD81 of cells treated with 0 vs. 10 μg/mL of PCSK9 for 3 h at 37 °C as assessed by confocal microscopy. The images provided are representative of two independent experiments. Scale bars represent 10 μm. **c** Cell viability of cells treated with PCSK9 relative to non-treated cells. $N = 2$ independent experiments. **d** Level of transduction or infection of cells treated with 0 vs. 10 μg/mL of PCSK9 for 3 h at 37 °C. The media was replaced 3 h p.i. or p.t. and the cells were harvested at 48 h p.i. or p.t. for determination of the levels of transduction or infection by flow cytometry. Independent experiments: $N = 5$ CCHF tecVLP; $N = 3$ HAZV, VSVpp and MLVpp. Two-way ANOVA test with Sidak's multiple comparisons ($H_2O$ vs. PCSK9: CCHF tecVLPs $p < 0.0001$, HAZV $p = 0.9736$, VSVpp $p = 0.0023$, MLVpp $p = 0.5648$). **e** Cell surface

expression of LDL-R of cells treated with 0 vs. 75 μM BBM for 2 h at 37 °C. Control isotypes are depicted in dotted lines. Representative image of 3 independent experiment. **f** Cell surface expression of LDL-R or CD81 of cells treated with 0 vs. 75 μM BBM for 2 h at 37 °C as assessed by confocal microscopy. The images provided are representative of two independent experiments. Scale bars represent 10 μm. **g** Cell viability of cells treated with 0 vs. 75 μM BBM for 2 h at 37 °C. $N = 4$ independent experiments. **h** Level of transduction or infection of cells treated with 0 vs. 75 μM BBM for 2 h at 37 °C. The media was replaced 3 h p.i. or p.t. and the cells were harvested at 48 h p.i. or p.t. for determination of the levels of transduction or infection by flow cytometry. Independent experiments: $N = 6$ CCHF tecVLP, $N = 3$ HAZV and MLVpp, $N = 4$ VSVpp. Two-way ANOVA test with Sidak's multiple comparisons (DMSO vs. BBM: CCHF tecVLPs $p < 0.0001$, HAZV $p = 0.0286$, VSVpp $p = 0.0006$, MLVpp $p = 0.8694$). Data are represented as means ± SEM.

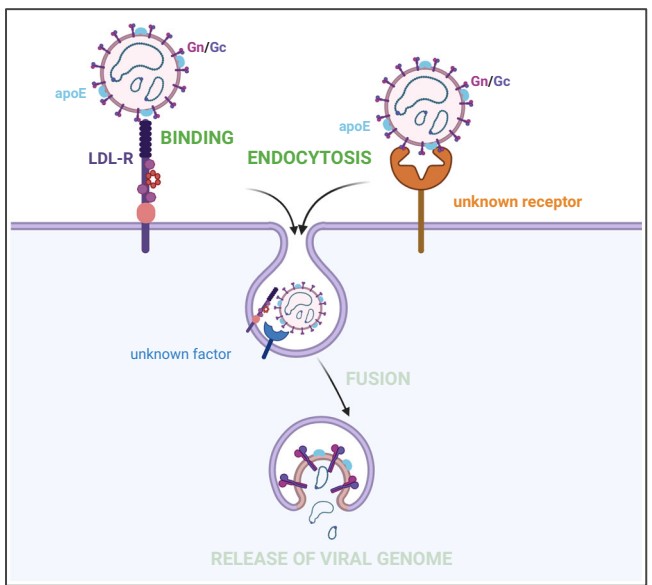

**Fig. 8 | Summary of the role of LDL-R in CCHFV entry.** CCHFV particles could incorporate apoE at their surface, which might contribute to the binding to LDL-R in addition to Gc. Followed this binding, CCHFV particles are endocytosed and then fuse with the late endosome membranes, allowing the release of viral genome. At this time, we cannot exclude the role of an additional factor (in blue) to promote endocytosis and/or fusion of CCHFV particles. In addition, an LDL-R-independent route of entry remains possible via a still unknown receptor (orange). Created with Biorender.com.

its surface GPs, which may promote assembly and secretion of viral particles in a manner reminiscent of HCV[55–57]. On the other hand, but not exclusively, as apoE is an exchangeable apolipoprotein[58,59], its incorporation on the lipid bilayer of viral particles may occur passively during their assembly in the Golgi or other organelles of the secretory pathway[60]. Finally, like for HCV, for which apoE association to viral particles could also occur in the extracellular environment[32] as well as in the vicinity of apoE-secreting target cells[33], CCHFV may recruit actively or passively apoE after virion egress.

CCHFV is detected in different organs in vivo upon infection and can infect several different cell types in vitro[61], hence underscoring the need for ubiquitous cellular receptors and cofactors for cell entry. In this respect, the broad tissue distribution of LDL-R suggests that it may promote entry in a variety of CCHFV target cell types. CCHFV infection is not restricted to humans as it can infect a large diversity of mammals such as cattle, sheep, goats, rhinoceroses, and camels[62], but this infection might not depend on LDL-R, at least according to our results with bovine cells. Interestingly, it seems that infection of CCHFV in mice is dependent on LDL-R[36], suggesting that its entry mechanism might differ between hosts. As above-stated, this could be due to

species-specific determinants, and we could speculate a low recognition of human apoE by bovine LDL-R or a preferential usage of another LDL-R family member in non-human cells. Likewise, CCHFV also replicates in tick cells, which poses the question of species-specific entry factors *vs.* receptors conserved across arthropod and mammal species. Interestingly, the vitellogenin receptor (VgR), which is expressed in arthropod oocytes, shares the similar architecture and functions of human LDL-R[63]. We note that a plant virus was shown to bind vitellogenin in order to mediate cell entry via VgR in its insect host, the true bug[64]. In this respect, it would be interesting to know if CCHFV uses a similar mechanism in tick cells. Finally, inter-human transmission of CCHFV has been reported[3], including nosocomial infections[65]. In that context, as incorporation of apoE on viral particles enhances infection through LDL-R (our report), apoE might play a crucial role in tropism and/or reservoir establishment in a manner dependent on tissue-specificity of apoE expression, such as in hepatocytes.

Taken together, our data identify a cellular receptor of CCHFV and its ligand incorporated within viral particles, highlighting a new and original mechanism developed by an important pathogenic bunyavirus.

## Methods
### Cells
Huh-7.5 cells (kind gift from Charles Rice), HEK-293T kidney cells (ATCC CRL-3216), TE-671 cells (ATCC CRL8805), A549 cells (kind gift from P. Boulanger), VeroE6 cells (ATCC CRL-1586), EBL cells (kind gift from Fabienne Archer), MDBK cells (European Collection of Authenticated Cell Cultures (ECACC)) were grown Dulbecco's modified minimal essential medium (DMEM, Invitrogen, France) supplemented with 100 U/mL of penicillin, 100 μg/mL of streptomycin and 10%. of fetal bovine serum.

PHH (BD Biosciences) were centrifuged in F12-HAM medium (Sigma Aldrich) and seeded overnight in collagen-coated plates in BD Gentest seeding medium supplemented with 5% FCS. 16 h later, PHH were washed and cultured with a culture medium for PHH (DMEM F12, Sigma Aldrich) supplemented with 10% FCS, 1 μg/mL BSA, 5 μg/mL bovine insulin, $1 \times 10^{-6}$ M Dexamethasone (Sigma Aldrich), $1 \times 10^{-8}$ M 3.3 trilodo-L-thyronin, 5 μg/mL apotransferrin, 1% of non-essential amino acids (Gibco), 1% of Glutamin (Gibco) and 1% Penicillin-Streptomycin solution (Gibco).

### Plasmids
The constructs encoding wild-type CCHFV strain IbAr10200 L polymerase (pCAGGS-V5-L), N nucleoprotein (pCAGGS-NP), M segment (pCAGGS-M), T7 RNA polymerase (pCAGGS-T7), nanoluciferase (nanoLuc)-expressing minigenome flanked by L NCR under the control of a T7 promotor (pSMART-LCK_L-Luc), GFP expressing minigenome flanked by L 5' and 3' UTRs under the control of a T7 promotor (pT7_vL_GFP), and an empty vector (pCAGGS) were described

previously[19,66] (all kind gifts from Friedemann Weber and Eric Bergeron). psPAX2, phCMV-G (kind gifts from Didier Trono and Jane Burns, respectively), and phCMV_HIV_GFP were used for VSV pseudo-particles production. psPAX2, phCMV_HIV_GFP and phCMV-4070A[67] were used for MLV pseudparticle production. pFK-JFH1/J6/C-846_Δp7, constructed from pFK-JFH1/J6/C-846 by deletion of p7 and addition of EMCV IRES between E2 and NS2, and phCMV-noSPp7(J6) were used for HCVtcp production. pMK-RQ-HAZV resQ S EGFP P2A, pMK-RQ-HAZV M, pMK-RQ-HAZV L (kind gift from John N. Barr) were used for production of rescued HAZV rHAZV-eGFP. phCMV-G, pCAGGS-M-Δ10, encoding for a CCHFV glycoprotein precursor lacking the last 53 residues of Gc[6], or with phCMV-EboV[68], encoding for EBOV glycoprotein were used for production of pseudotyped VSV-ΔG. For VDL-R and apoE expression, pCSII-EF-VLDLR-HA (kind gift from Yoshiharu Matsuura) and pWPI-hApoE3 (kind gift from Ralf Bartenschlager) respectively were used for production of lentivirus in combination with phCMV-G and psPAX2. For the down-regulation assays, TRC2_pLKO_shLRP1 (TRCN0000257100; Sigma-Aldrich), TRC2_pLKO_shLDL-R (TRCN0000262146; Sigma-Aldrich); TRC1_pLKO_shapoE (TRCN0000010913; Sigma-Aldrich) or plasmids described in[69] or pHR-SIN-CSGW (empty backbone) were used for generation of lentivirus in combination with phCMV-G and psPAX2.

## Antibodies
The list of all antibodies used in this study is available in the Supplementary information (Supplementary Table 1).

## Production of viral stocks and infection assays with WT CCHFV particles
All the experiments with WT live CCHFV were performed in the Jean Mérieux BSL-4 facility in Lyon, France. To produce viral stocks, Huh-7.5 cells were infected using CCHFV isolate IbAr10200 (obtained from Institut Pasteur) at MOI of 0.01 and the production was harvested at 72 h post-infection. Infectious titers were determined by NP immunostaining on VeroE6 cells[70] using anti-NP (2B11) as primary antibody and viral preparations with titers ranging between $3 \times 10^5$ $10^6$ NP FFU/ml were used in this study.

For blocking and neutralization assays, infections were performed with serial dilutions of viral stocks, corresponding to MOIs of 0.5 to 0.001. Viral stocks or cells were treated as described below. 24 h post-infection, infected cells were lysed with TRIzol™(ThermoFisher), allowing inactivation of virus, and RNAs were extracted according to manufacturer's protocol and level of viral RNA, reflecting the level of infection, was determined by RT-qPCR (see below). The viral titer was determined after selection of dilutions allowing a linear range of viral RNA signal.

## Production of viral stock and transduction assays with CCHF tecVLPs
For production of tecVLPs from IbAr10200 CCHFV strain (Fig. 1a), Huh-7.5 or HEK-293T cells were transfected in 10 cm dishes with 3.6 μg of pCAGGS-V5-L, 1.2 μg of pCAGGS-NP, 3 μg of pCAGGS-M or pCAGGS, 3 μg of pCAGGS-T7 and 1.2 μg of pSMART-LCK_L-Luc (for tecVLP-NanoLuc) or pT7-GFP (for tecVLP-GFP), using GeneJammer transfection reagent (Agilent). 6 hours post-transfection, cells were washed two times with OptiMEM before addition of OptiMEM. At 72 h post-transfection, supernatant was harvested and filtered through a 0.45 μm filter. Preparations of tecVLPs with titers of $5 \times 10^5$ GFP transduction units (t.u.)/ml (for tecVLP-GFP) or $10^8$ RLU/ml (for tecVLP-NanoLuc) were used in this study.

For assays with tecVLP-GFP (Fig. 1a), targets cells were pre-transfected using 2.4 μg of pCAGGS-V5-L and 4.8 μg of pCAGGS-NP using GeneJammer transfection reagent. 6 hours post-transfection, cells were seeded in 24, 48 or 96-well plates in OptiMEM. 24 h post-transfection, cells were transduced with serial dilutions of particles, corresponding to MOIs of 2 to 0.02, and 48 h post-transduction, transduced cells were harvested. For each dilution, technical replicates were performed. Transduced cells were fixed and the percentage of GFP positives cells was assessed by flow cytometry (MACSQuant® VYB Flow Cytometer; Miltenyi Biotec). Data were analyzed with FlowJo software (BD Biosciences). The viral titer was determined after selection of dilutions within the linear range of percentages of positive cells.

For assays with tecVLPs with a nanoLuc minigenome (Fig. 1a), the transduction with serial dilutions of viral supernatant, corresponding to RLU-per-cell of 100 to 0.01, was done on Huh-7.5 cells stably expressing firefly luciferase (FLuc) and the level of transduction was quantified 24 h post-transduction, by lysing the cells with passive lysis buffer (Promega) for 10 min at room temperature and measurement of luciferase signal using Nano-Glo® Dual-Luciferase® Reporter Assay System (Promega). For each dilution, technical replicates were performed. The viral titer was determined after selection of dilutions within a linear range of nanoLuc signals.

## Production and infection assays with HAZV particles
For production of viral stocks, rHAZV-eGFP virus[24] was amplified in Huh-7.5 cells (MOI = 0.001). 1 h post-infection, media was changed after a PBS wash and 72 h post-infection, supernatant was harvested and clarified by centrifugation 5 min at 750 x g. Preparations of rHAZV-eGFP (termed HAZV in the text and figures) with titers of $10^6$ eGFP infection units (i.u.)/ml were used in this study.

For infection assays, Huh-7.5 cells were inoculated with serial dilutions of viral supernatant, corresponding to MOIs of 0.5 to 0.001, before PBS wash and medium change, 1 h post-infection. Level of infection was detected 16 h post-infection by quantification of eGFP positive cells by flow cytometry (MACSQuant® VYB Flow Cytometer; Miltenyi Biotec). Data were analyzed with FlowJo software (BD Biosciences). The viral titer was determined after selection of dilutions allowing a linear range of percentage of positive cells.

## Production and infection assays with HCV trans-complemented particles (HCVtcp)
For production of viral stocks, Huh-7.5 cells were electroporated with 2 μg of phCMV-noSPp7 DNA and 10 μg of Jc1 Δp7 in vitro transcribed RNA as described previously[71]. Media was changed 6 h post-electroporation and supernatant was harvested and filtered (0.45 μm) 72 h later. Preparations of HCVtcp with titers of $10^3$ NS5A FFU/ml were used in this study.

For infection assays, Huh-7.5 cells were inoculated with serial dilutions of viral supernatant, corresponding to MOIs of 2 to 0.02, and were fixed using ethanol 48 h post-infection and focus-forming units were determined by counting NS5A immunostained foci. The viral titer was determined after selection of dilutions allowing a linear range of foci.

## Production and transduction assays with VSV or MLV pseudoparticles
Lentiviral vectors encoding GFP sequence and bearing VSV-G (VSVpp) or amphotropic MLV Env glycoprotein (MLVpp) were produced in HEK-293T cells by transfection of phCMV_HIV_GFP, psPAX2 and phCMV-G or phCMV-4070A using calcium phosphate precipitation. Media was replaced 16 h later and supernatant was harvested and filtered (0.45 μm) 24 h later. Preparations of VSVpp and MLVpp with titers of $2 \times 10^6$ and $6 \times 10^5$ GFP t.u./ml, respectively, were used in this study.

For transduction assays, Huh-7.5 cells were transduced with serial dilutions of viral supernatants, corresponding to MOIs of 2 to 0.02, and were fixed 48 h post-transduction. The percentage of GFP-positive cells was assessed by flow cytometry (MACSQuant® VYB Flow Cytometer; Miltenyi Biotec). Data were analyzed with FlowJo software (BD Biosciences). The viral titer was determined after selection of dilutions allowing a linear range of percentage of positive cells.

## Production and transduction assays with VSV-ΔG

HEK-293T cells were transfected with phCMV-G, or with pCAGGS-M-Δ10, encoding for a CCHFV glycoprotein precursor lacking the last 53 residues of Gc[6], or with phCMV-EboV[68] encoding for EBOV glycoprotein. At 24 h post-transfection, cells were transduced with VSV-ΔG/GFP*G, encoding for a GFP instead of G protein at MOI = 5. After 1 h, medium was replaced by OptiMEM in presence of anti-VSV-G 41A1. Cell supernatant was harvested 16 h post-transduction. Preparations of VSV-ΔG/GFP*G, VSV-ΔG/GFP*CCHFV, and VSV-ΔG/GFP*EBOV with titers of $2 \times 10^8$, $7 \times 10^5$ and $1.7 \times 10^5$ GFP t.u./ml, respectively, were used in this study.

For transduction assays, Huh-7.5 cells were transduced with serial dilutions of viral supernatants, corresponding to MOIs of 2 to 0.02, and were fixed 24 h post-transduction. The percentage of GFP-positive cells was assessed by flow cytometry (MACSQuant® VYB Flow Cytometer; Miltenyi Biotec). Data were analyzed with FlowJo software (BD Biosciences). The viral titer was determined after selection of dilutions allowing a linear range of percentage of positive cells.

## Down-regulation of lipid receptors

Lentiviral vectors expressing shRNA targeting LRP1, LDL-R, SRBI or control shRNA were produced in HEK-293T cells. Huh-7.5 cells stably expressing Firefly Luciferase were transduced with lentiviral vectors at MOI = 30. Four days later, cells were transduced with serial dilutions of tecVLPs with the nanoLuc minigenome. The knock-down was assessed by western blot of cell lysate generated 4 days post-transduction and using anti-LRP1, anti-LDL-R and anti-CD36L1. Level of transduction was assessed 24 h post-transduction with measurement of luciferase signals using Nano-Glo® Dual-Luciferase® Reporter Assay System (Promega) as described above.

## Down-regulation of apoE

Lentiviral vectors expressing shRNA targeting apoE or control shRNA were produced in HEK-293T cells. Huh-7.5 cells were transduced with lentiviral vector. 24 h post-transduction, cells were transfected with plasmid allowing tecVLPs production or infection of HAZV. 72 h post-transfection, supernatants were harvested and used for assessment of transduction efficiency or infectivity and RNA levels. Level of KD was checked by apoE intracellular FACS staining. Cells were fixed and permeabilized with Cytofix/CytoPerm (BD Biosciences) according to manufacturer instructions. Cells were incubated with primary antibody (AHP2177, 1/2000) diluted in Perm/Wash buffer (BD Biosciences) for 1 h at 4 °C with regular checking. After three washes with Perm/Wash buffer, cells were incubated for 1 h at 4 °C with secondary antibody. Cells were washed three times with Perm/Wash buffer before resuspension in PBS and flow cytometry acquisition (MACSQuant® VYB Flow Cytometer; Miltenyi Biotec). Data were analyzed with FlowJo software (BD Biosciences). The gating strategy is depicted in the Supplementary Fig. 4.

## Overexpression of VLDL-R

Huh-7.5 cells stably expressing Firefly luciferase, were transduced at MOI = 30 with a lentiviral vector encoding VLDL-R and transduced at 4 days post-transduction with serial dilutions of tecVLPs containing a nanoLuc expressing minigenome. The level of VLDL-R at the cell surface was assessed by flow cytometry at the day of transduction. Level of transduction was assessed 24 h post-transduction with measurement of luciferase signals using Nano-Glo® Dual-Luciferase® Reporter Assay System (Promega) as described above.

## Overexpression of apoE

HEK-293T cells were transduced with a lentiviral vector encoding apoE and were then cultivated in presence of blasticidin for selection of transduced cells. Selected cells were used for production of tecVLPs as described above. The intracellular level of apoE was assessed by flow cytometry as described for down-regulation.

## Blocking with anti-LDL-R antibody

All cell lines were grown in OptiMEM and were incubated with different doses of anti-LDL-R or control IgG for 1 h at 37 °C. Then viral inoculum was added to cells in presence of antibodies, and media was replaced with DMEM, 10% FCS, 3 h post-transduction/infection. All experiments were performed with serial dilution of viral supernatants. For tecVLP-GFP or lentiviral pseudoparticles, cells were harvested 48 h post-transduction and level of transduction was determined by flow cytometry and titer was obtained as described above; for WT virus, cells were harvested 24 h post-infection and infectious titer was determined by RT-qPCR as described above; for HAZV, cells were harvested 16 h post-infection and infectious titer was determined by flow cytometry as described above. When testing different cell lines (EBL, MDBK, A549, TE-671, HEK-293T), cells were transduced with tecVLP-NanoLuc, the level of transduction was quantified 48 h post-transduction, by lysing the cells with passive lysis buffer (Promega) and measurement of luciferase signal using Nano-Glo® Luciferase Assay System (Promega). Viral titer was determined as described above. For analysis of incubation kinetics, Huh-7.5 cells were incubated with LDL-R antibody either 1 h before transduction, at the time of transduction or 2 h, 4 h, 6 h post-transduction. The antibody-containing media was replaced by fresh at 2 h or at 24 h post-transduction depending on the conditions (Fig. 3a). Cells were harvested at 48 h post-transduction, and the viral titer was determined after the detection of positive cells by flow cytometry as described above.

For transduction of PHH, at 24 h post-seeding, cells were washed and incubated in their culture medium, with different doses of anti-LDL-R or control IgG for 1 h at 37 °C before transduction with serial volumes of tecVLP-NanoLuc in presence of antibodies. 3 h post-transduction, medium was changed, and the level of transduction was assessed 24 h post-transduction, as described above.

## Neutralization assays with sLDL-R or apoE antibodies

Serial dilutions of inoculate were incubated for 1 h at room temperature with different doses soluble LDL-R (sLDL-R), CD81_6His_LEL, anti-apoE serum or control goat serum and then added to Huh-7.5 cells grown in OptiMEM. All the infection/transduction assays were performed with serial dilutions. At 3 h post-infection or -transduction, media was replaced with DMEM, 10% FCS. For tecVLP-GFP or lentiviral pseudoparticles, cells were harvested 48 h post-transduction and level of transduction was determined by flow cytometry and viral titer was determined as described above; for WT CCHFV virus, cells were harvested 24 h post-infection and level of infectivity was determined by RT-qPCR as described above. For HAZV, infected cells were harvested 16 h post-infection and level of infectivity was determined by flow cytometry. For HCV, cells were fixed 48 h post-infection and level of infectivity was determined by immunostaining as described above.

## Binding assays

CCHF tecVLPs were incubated with 5 μg of sLDL-R or CD81-LEL[72] (both harboring a 6xHis tag) for 1 h at room temperature before incubation with Ni-particles (MagneHis™ Protein Purification System, Promega), according to the manufacturer's protocol. After 3 washes, beads were resuspended in TriReagent before extraction and determination of the level of cocaptured CCHFV minigenome by RT-qPCR.

## Cell surface staining of LDL-R or VLDL-R

For flow cytometry, cells were washed and detached with Versene (Invitrogen), before fixation with 2% paraformaldehyde (PFA, (Sigma-Aldrich, France)) for 15 min. Cells were then incubated for 1 h at 4 °C with primary antibody anti-LDL-R (AF2148) or control isotype at 40 μg/mL in PBS + 2% FCS, with regular shaking. After 3 washes, cells

were incubated with secondary antibody for 1 h at 4 °C in PBS + 2% FCS, with regular shaking. After 3 washes, cells were resuspended in PBS + 2% FCS and analyzed by flow cytometry (MACSQuant® VYB Flow Cytometer; Miltenyi Biotec). The gating strategy is depicted in the Supplementary Fig. 3. For VLDL-R staining, the same protocol was used using anti-VLDLR (1H10).

For cell imaging, Huh7.5 cells were grown in 6 well plates containing coverslips. 24 h later, cells were treated as indicated in the figure. Treated cells were then fixed with 3% PFA for 15 min and directly processed for immuno-staining after 3 washes with PBS. Fixed cells were saturated with 3% bovine serum albumin (BSA)/PBS for 20 min and incubated for 1 h with anti-LDL-R (AF2148, 20μg/mL) and anti-CD81 (JS-81, BD Pharmingen, 1/250) diluted in 1% BSA/PBS. After three washes with 1% BSA/ PBS, cells were incubated for 1 h with donkey anti-goat Alexa Fluor 488 and donkey anti-mouse Alexa Fluor 555 respectively (A-11055 and A-31570 respectively, Molecular Probes) at a 1/2000 dilution in 1% BSA/PBS. Cells were washed three times with PBS, stained for nuclei with Hoechst 33342 (H3570, Molecular Probes) for 5 min in PBS, washed three times with PBS and mounted in Mowiol 40-88 (Sigma-Aldrich) before acquisition with confocal microscope LSM-800 (Zeiss) equipped with a 63X objective. Images were analyzed with the Fiji software (https://imagej.net).

## Co-immunoprecipitation assay

CCHF tecVLPs were incubated with apoE antibodies (AB947; Sigma-Aldrich) or control goat IgG overnight at 4 °C. Then 1.5 mg of Dyna-beads protein G magnetic beads (ThermoFisher) were added during 1 h at room temperature. The beads were then washed 3 times with PBS. For the elution, beads were resuspended in TriReagent and the supernatant was transferred into a new tube for RNA extraction, following manufacturer's protocol before determination of the level of cocaptured CCHFV minigenome by RT-qPCR (see below).

The same procedure was used with CCHFV WT particles, with elution using Trizol LS. RNA and proteins were extracted according to the manufacturer's protocol. The level of co-captured CCHFV RNA was determined by RT-qPCR (see below) and the level of proteins was determined by western blot.

## Capture of particles with sLDL-R or CD81-LEL

CCHF tecVLPs or WT CCHFV or HAZV particles were incubated with 5μg of soluble LDL-R (2148-LD-025/CF, R&D systems), or CD81-LEL for 1 h at room temperature. Then 30uL of MagneHis Ni-Particles (Promega) were added and the samples were processed according to manufacturer's instructions. For the elution, beads were resuspended in TriReagent and the supernatant was transferred into a new tube for RNA extraction, following manufacturer's protocol before determination of the level of co-captured CCHFV or HAZV RNAs by RT-qPCR (see below).

## Detection of viral genomes by RT-qPCR

After extraction following the manufacturer's protocol, RNA was reverse transcribed (iScript cDNA synthesis kit; Bio-Rad). In the case of tecVLPs samples, RNA was treated with DNAse (Invitrogen) according to manufacturer's protocol. Level of cDNA was then quantified by qPCR. For tecVLPs minigenome, the quantification was done by detection of the nanoLuc minigenome for CCHFV: 5′-TAGTCGAT-CATGTTCGGCGT-3′ and 5′-ACCCTGTGGATGATCATCACT-3′ with 5′-GATTACCAGTGTGCCATAGTGCAGGATCAC-3′ as probe, using TaqMan™ Gene Expression Master Mix (ThermoFisher). For WT CCHFV, the quantification was done using FastStart Universal SYBR (Roche) with the following primers 5′-CCCCACACCCCAAGATAATA-3′ and 5′-ACTACTCTGCATTCTCCTCA-3′ targeting L UTR. For HAZV, the quantification was done by using FastStart Universal SYBR (Roche) with the following primers 5′-CAAGGCAAGCATTGCACAAC-3′ and 5′-GCTTTCTCTCACCCCTTTTAGGA-3′ targeting S segment.

For titration of WT CCHFV, viral RNA levels were normalized with respect to glyceraldehyde-3-phosphate dehydrogenase (GAPDH) RNA levels, detected using FastStart Universal SYBR (Roche) and specific primers 5′-AGGTGAAGGTCGGAGTCAACG-3′ and 5′-TGGAA-GATGGTGATGGGATTTC-3′.

As an internal control of extraction, an exogenous RNA from the linearized Triplescript plasmid pTRI-Xef (Invitrogen) was added into the supernatant prior to extraction and quantified with specific primers (5′-CGACGTTGTCACCGGGCACG and 5′-ACCAGGCATGGTGGT-TACCTTTGC). This signal was used for normalization of signal for crude supernatant, as well as capture and IP assays.

All analyses were done on a Quantstudio real-time PCR apparatus.

## Drug treatment

Recombinant PCSK9 (Thermo-Fisher) or Berbamine (BBM, Sigma-Aldrich) were used at the indicated concentration and time of incubation. Huh-7.5 cells were incubated with 0 vs. 10μg/mL of PCSK9 for 3 h at 37 °C, or with 0 vs. 75μM of Berbamine (BBM) for 2 h at 37 °C before infection or transduction with serial volume of particles. The media was replaced 3 h post-infection (p.i.) or post-transduction (p.t.) and the cells were harvested at 48 h p.i. or p.t. for determination of the levels of infection or transduction by flow cytometry. Cell viability and level of LDL-R at cell surface were assessed at the time of infection or transduction.

## Electron microscopy

For negative staining, formvar/carbon-coated nickel grids were deposited on a drop of samples during five minutes and rinsed two times on drop of water. The negative staining was then performed with three consecutive contrasting steps using 2% uranyl acetate (Agar Scientific), before analysis under the transmission electron microscope (JEM-1400Plus).

For analysis of the particles by negative staining electron microscopy and immunogold labeling, formvar/carbon-coated nickel grids were deposited on a drop of samples during five minutes and rinsed two times with phosphate-buffered saline (PBS). Grids were then incubated on a drop of PBS/BSA 1% and then PBS containing 1:100 goat anti-apoE serum (AHP2177) and 1:100 mouse anti-Gn serum (in-house) for one hour. After six washes of five minutes with PBS, each, grids were further incubated for one hour on a drop of PBS containing 1:30 gold-conjugated (6 nm) donkey anti-goat or 1:30 gold conjugated (6 nm) goat anti-mouse (Aurion). Grids were then washed again with six drops of PBS, post-fixed in 1% glutaraldehyde and rinsed with three drops of distilled water. The negative staining was then performed with three consecutive contrasting steps using 2% uranyl acetate (Agar Scientific), before analysis under the transmission electron microscope (JEM-1400Plus).

## Western blot analysis of cell lysates and pellets

For cell lysates, cells were lysed with lysis buffer (20 mM Tris [pH 7.5], 1% Triton X-100, 0.05% sodium dodecyl sulfate, 150 mM NaCl, 0.5% Na deoxycholate)) supplemented with protease/phosphatase inhibitor cocktail (Roche) and clarified from the nuclei by centrifugation at $13,000 \times g$ for 10 min at 4 °C for quantitative western blot analysis. For pelleting of particles, supernatants were harvested and filtered through a 0.45μm filter and centrifuged through a 20% sucrose cushion at 28,000 rpm for 2 h at 4 °C with a SW41 rotor and Optima L-90 centrifuge (Beckman). Pellets were resuspended in PBS prior to use for western blot.

Proteins obtained in total cell lysates or pellets were denatured in Laemmli buffer (250 mM Tris-HCL pH 6.8, 10% SDS, 50% glycerol, 500 mM β-mercapto-ethanol, bromophenol blue) at 95 °C for 5 min separated by SDS-PAGE, and then transferred to nitrocellulose membrane and revealed with specific primary antibodies, followed by the addition of IRdye secondary antibodies, and imaging with an Odyssey

infrared imaging system CLx (Li-Cor Biosciences). In the case of Gc detection, proteins in total cell lysates or pellets were loaded in non-denaturing, non-reducing buffer (250 mM Tris-HCL pH 6.8, 5% SDS, 50% glycerol, bromophenol blue).

## Cell viability measurement

The cell viability was assessed using Cytotox-Glo Cytotoxicity Assay (Promega) according to the manufacturer's protocol.

## Statistical analysis

Significance values were calculated by applying tests indicated in the figure legends using the GraphPad Prism 10 software (GraphPad Software, USA). P values under 0.05 were considered statistically significant and the following denotations were used: ****$P < 0.0001$; ***$P < 0.001$; **$P < 0.01$; *$P < 0.05$; ns (not significant), $P > 0.05$.

## Reporting summary

Further information on research design is available in the Nature Portfolio Reporting Summary linked to this article.

## Data availability

The data generated in this study are provided in the Source Data file. Source data are provided with this paper.

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

## Acknowledgements

We thank Chloé Journo and Christiane Riedel for stimulating discussions. We thank Fouzia Amirache, Christelle Granier, Solène Lerolle, Chloé Mialon and Johan Toesca for their technical inputs. We are grateful to Ralf Bartenschlager for the gift of HCV and apoE constructs, Charles Rice for the Huh-7.5 cells and 9E10 NS5A monoclonal antibody, Yoshiharu Matsuura for the VLDL-R construct, Friedemann Weber and Eric Bergeron for the CCHF tecVLP constructs, Olivier Reynard for the VSV-ΔG/GFP stock, and Sandra Lacôte for the bovine cell lines. We acknowledge the contribution of SFR Biosciences (Université Claude Bernard Lyon 1, CNRS UAR3444, Inserm US8, ENS de Lyon), LYMIC-PLATIM-microscopie, AniRA-Cytométrie and AniRA vectorology, especially Caroline Costa. We are grateful to P4 Jean Mérieux team (INSERM

US03, Lyon) and the related biosafety team for their assistance for BSL4 activities. This work was supported by the LabEx Ecofect (ANR-11-LABX-0048 awarded to F.-L.C.) of the Université de Lyon, within the program Investissements d'Avenir (ANR-11-IDEX-0007) operated by the French National Research Agency (ANR), the Fondation pour la Recherche Médicale (FRM, Grant number: EQU202203014673 awarded to F.-L.C.), the ANRS I MIE (Grant number: ANRS0630 awarded to F.-L.C.), and the ANR (Grant number: ANR-22-ASTR-0031 awarded to F.-L.C.). S.D. and A.Gau. were supported by fellowships of the ANRS I MIE.

## Author contributions

Conceptualization, M.R., F.-L.C., and S.D.; Investigation, M.R., L.C., A.Gau., T.V., L.Z., A.L., B.B., A.Gan., S.B., J.B.G., N.F., V.Le., C.M.; Writing Original-Draft, M.R., F.-L.C., and S.D.; Writing Revised-Draft, F.-L.C. and S.D.; Supervision, F.-L.C and S.D.; Critical reagents, P.R., J.N.B., V.Lo.; Funding Acquisition, V.Lo., V.Le., F.-L.C. and S.D.

## Competing interests

The authors declare no competing interests. A European patent application has been filed by Inserm-Transfert.
