## [Peer Review File · Nature Communications]

The low-density lipoprotein receptor and apolipoprotein E associated with CCHFV particles mediate CCHFV entry into cellsREVIEWER COMMENTS

Reviewer #1 (Remarks to the Author):

Overview: LDLR family members have recently been shown to mediate cellular entry by diverse viruses. Specifically, Lrp1 was shown to be a major host cell entry factor for Rift valley fever virus (RVFV) and Oropouche virus (OROV), both members of Bunyavirales order. Here, Ritter et al present data that the low-density lipoprotein receptor (LDL-R) mediates binding and internalization of Crimean-Congo hemorrhagic fever virus (CCHFV) VLPs and, to a lesser degree, authentic CCHFV, in human hepatocytes but not bovine cells. Interestingly, binding to LDLR does not appear to be due to direct binding by CCHFV GP, but rather binding is mediated by apoE expressed in the viral membrane obtained from the producer cell. This is a new and intriguing mechanism of virus-co-opted entry mechanisms. While the data presented are interesting and exciting, the overall manuscript suffers from data presentation that is confusing and often difficult to interpret, as detailed below. In addition, the majority of the experiments are performed with tecVLPs, where a strong LDLR-dependent phenotype is observed, while only a few experiments are done with authentic CCHFV, and the LDLR-dependent phenotype is less significant. More detailed biochemical (i.e. direct binding between CCHFV GP and LDLR, or lack thereof) and virological (additional studies with wt CCHFV) experiments should be considered to support the authors conclusions. Future in vivo studies to validate these findings should also be considered in the discussion, although it is likely outside the scope of this manuscript. While this manuscript is a fascinating contribution to the field as is, biological significance of these findings would be underscored with additional biochemical, virological, and in vivo experimentation to highlight CCHFV dependence on LDL-R and ApoE.

Major Comments:

1. Explanation of the tecVLP system - what they are, how they are made, and why they are a suitable substitute for authentic virus - needs to be detailed in the manuscript. A diagram or figure may be helpful here. Limitations of this system need to be discussed also.
2. Many of the inhibition assays throughout all figures show the data in a counterintuitive manner - i.e. the y-axis is % of inhibition with 0 on the bottom and inhibition rises. Intuitive presentation would be to graph the data as a % of control, starting at 100% and going down.
3. The strong blocking of tecVLPs with anti-LDLR mAb seen in Fig 1C is not replicated in Fig 1D with authentic CCHFV. Same with Fig 2C, 2D, 3A, 3B. For authentic CCHFV studies, additional replicates at each dose should be performed, and actual viral titers should be included for reference. Discussion of the discrepancy between the tecVLP and CCHFV data should be included in the discussion.
4. Figure 1C and 1D: were there biological replicates in addition to the three independent technical replicates? Are these datapoints an average or only an n=1 from each experiment? This should be clarified if an average of three replicates per experiment replicate are plotted or additional replicates should be performed and those datapoints should be plotted.
5. Key experimental details are lacking in the Results section which makes interpretation of the data more difficult. Examples:
 - a) Lines 79-81: What method was used to knock down the lipid transfer receptors?
 - b) Lines 84-86: For tecVLP assays here and elsewhere, critical experimental information such as MOI, timepoint harvested, and experimental readout (i.e. luciferase assay vs flow cytometry vs vRNA) need to be included in the results.
 - c) Lines 94-96: For CCHFV infection studies, experimental information such as stock/strain used, MOI, time point, readout, and actual titers should be included.
 - d) Figs 1C and 2C. Y-axis is % inhibition compared to control, and infectivity (as stated in the legend) is measured by flow cytometry for GFP. This type of experimental details need to be included in the results for proper interpretation of the data. How is GFP expression quantified? Is it % of cells expressing GFP? Or MFI of GFP? It is very unclear in the manner currently presented. Same question for Figs 1D and 2D - how is viral RNA quantified and converted to % inhibition of control?
 - e) Lines 143-147 - how was capture assay performed? Basic information should be included in the

Results.

f) Similar to the above comment, in Fig 2F, G, H, data is presented as "level of infection compared to control" (y-axis), and it is difficult to assess whether a change from 1.0 (ctl) to 0.8 (Tyr) is biologically meaningful despite statistical significance. What exactly is being measured (legend indicates GFP expression by flow cytometry) and how is it quantified and compared? If it is % of infection, like Fig 1F, it should be expressed as such. Very difficult to assess the biological relevance of the data on endocytosis in Fig 2 F, G, H.

1. Line 173-175 – the word "associate with" or "associate to" is used and the meaning is unclear. Sometimes it seems as if "associate" means "binds" as in Line 173-175 and in other places it may mean something different (lines 97-98).
2. Lines 206-207 states that results from Fig 3F indicate that "LDL-R dependent entry of CCHFV does not involve its surface glycoproteins" should be revised because the results indicate that LDLR-mediated CCHFV entry is dependent upon apoE.
3. Fig 4 would be bolstered by inclusion of fluorescence microscopy to visualize downregulation of surface expression of LDLR after treatment with PCSK9.

Minor comments:

1. Line 78: why were Huh 7.5 cells chosen for use and what LDLRs do they naturally express?
2. Fig 1B: y-axis is "Level of Infection (compared to control cells)" is unclear. Is this % infection?
3. Fig 1F: the orange/red/blue text legend on the left is unclear as to what it is referring to.
4. Fig 2B: graph is presented horizontally instead of vertically which makes it difficult to interpret compared to similar graphs throughout the manuscript. Suggest keeping the orientation consistent throughout.
5. Fig 3A,B: x-axis should indicate concentration of anti-apoE, rather than dilution
6. Fig 3G: This should not be a line graph and instead should be a bar graph with data points as in Fig 3A.
7. Line 86/243: It appears the shRNA downregulation of Lrp1 promoted infection of Huh7.5 cells. LDL-R family receptors often compensate for one another. Therefore the authors should test if shRNA KD of Lrp1 resulted in an increase in LDL-R surface expression which could then explain the increase in infectivity observed. Alternatively, KD of both or multiple LDL-R family receptors.
8. Line 99: since other viruses utilize LDL-R, it is not specifically only used by CCHFV. It may be more accurate to state "suggesting that LDL-R is not a pan-orthonairovirus entry factor"
9. Line 136: Similar to the use of HAZV in Figure 1, a negative control virus should be used in the sLDL-R neutralization assay (Figure 2C) to demonstrate that it does not have a pan anti-viral impact on Huh7.5 cells
10. In general, there is no comment on biosafety level of CCHFV, biological containment, and inactivation of samples. This reviewer believes this would add to the significance of investigating this virus. This is usually mentioned in the methods. In addition, no details are given on the q-RT-PCR assay used for CCHFV.
11. Line 206: Figure 4F should be Figure 3F
12. Line 211: Figure 4G should be Figure 3G
13. Line 226: Similar to the use of HAZV in Figure 1, another virus should be shown as a negative control to demonstrate that the slight decrease in viability (although not significant) following PCSK9 treatment does not impact general viral infection in a non-LDL-R manner

Editorial points to help with readability:

- Line 28: "cause" should read "can cause"
- Line 31: "as CCHFV" should read "as a CCHFV entry factor"
- Line 31: remove "could" and instead write "LDL-R promoted binding and internalization"
o This line in general is confusing: "LDL-R could promote binding and internalization of CCHFV particles" perhaps instead it should read "LDL-R facilitated binding and internalization..."
- Line 33: "mediate" should be "mediates the LDL/LDL-R"
- Line 34: "incorporated on CCHFV" should perhaps be "incorporated in CCHFV particles"
- Line 36: "hijacked" could instead be "utilizes" or "takes advantage of"
- Line 44: "i.e." should be removed
- Line 89: delete "of" and "by"
- Line 92: either define KD in line 79 or spell out
- Line 102: delete "the". It should read "with the exception of hepatocytes"
- Line 146: "ca.?" what does this mean. Perhaps spell out "approximately"?

- Line 151: should read "we determined the effect of TyrA23 on both" or "TyrA23's"
- Line 155: should read "impairs" instead of "impaired"
- Line 180: should read "As a positive control,..."
- Line 181: should read "as a negative control.."
- Line 203: should read "(Supplemental Figure 3A) but are fully able to..."
- Line 219: "decreases" should read "decreased"
- Line 220: readability would improve if it read "LDL-R expression at the cell surface"
- Line 224: "level" should read "levels"
- Line 261: should read "combined with our report"
- Line 267/268: delete "e.g."
- Line 274: should read "bind to a large variety of ligands"
- Line 284: "factor" instead of "factors"
- Line 288: citation formatting needs to be edited
- Line 298: missing a . between "membranes" and "In"
- Line 299: "raise" should be "raises"
- Line 323: underline should be removed
- Line 361: "R&D System" should read "systems"
- Line 364/365: anti should possibly be capitalized
- Line 386: there should be an "and" before "filtered"
- Line 416: "filtered" spelling error
- Line 421: "Firefly Luciferase cells"? do the authors mean "Huh7.5 cells expressing firefly luciferase were transduced..."
- Line 452: do authors mean "Inoculates were incubated"?
- Line 727 in Figure 1 legend: "stably expressing Fluc were transduced" ("cells" should be deleted)
- Line 746 in Figure 2 legend: "ON" should be clarified as "overnight"
- Line 763: "were treated with DMSO" ("for" should be deleted)
- Line 810/811 etc: symbols

Reviewer #2 (Remarks to the Author):

Ritter et al. report on the identification of low-density lipoprotein receptor as a cellular receptor for the tick-borne emerging pathogen Crimean-Congo hemorrhagic fever virus (CCHFV). They used a series of approaches including antibody blocking studies and blocking studies with soluble receptor fragments. They further attempted to show that the LDLR ligand, apoE, was incorporated into viral particles and was responsible for LDLR-mediated entry, although the latter was only demonstrated using some indirect assays (e.g., showing that tecVLPs have apoE when they are produced in huh7 cells, that antibody against apoE can block entry of tecVLPs and authentic virus, and that anti-apoE antibody can enrich CCHFV minigenome RNA in a pull-down assay). Overall, while the work implicating LDLR as a cellular entry factor is compelling, the findings implicating the role of ApoE as an intermediate factor that is required for LDLR engagement are less robust. Critical controls are lacking, and experiments exploring a direct interaction between the Gn/Gc envelope proteins with LDLR are also missing. Lastly, the data with fully infectious virus have a great deal of variability which makes the robustness of the findings hard to appreciate. Additional concerns in regards to the statistical analysis are listed below.

MAJOR:

1. Figure 1D is a critical experiment attempting to confirm the results of antibody-blocking studies performed with surrogate systems (e.g., tecVLPs) but with authentic virus. The variability between the data points is quite high (from 0 to 70%). Could the authors comment on the likely source of variability?
2. Figure 2D, which includes soluble LDLR blocking assays with authentic virus – again, like with the antibody blocking studies, the variability in these experiments is quite high. I have the same

points to raise as those noted above.

3. Figure 2E. Is the capture of viral RNA by CD81-LEL nonspecific? It seems like a "beads alone" control is missing in this experiment (e.g., in this case, based on the methods, this would be "Niparticles" alone).

4. Lines 104-105: "Interestingly, expression of VLDL-R increased infection levels of tecVLPs by 2-3 folds (Figure 1F)" It looks like the change may not be statistically significant ($p=0.0625$) - this should be noted also in the main text.

5. Statistical analyses should be made between control and treatment, not just across concentration gradients. In Fig. 1C-E, 2B, 3A-B, 3F: please show comparison between anti-LDLR treatment and control IgG.

6. Fig. 3A and 3B - Control IgG should be included in these experiments.

7. Lines 166-168: "This treatment prevented both LDL-R endocytosis (Supplemental Figure 2D) and tecVLP infection (Figure 2H), suggesting that LDL-R is involved in CCHFV endocytosis." The p value in Figure 2H comparing DMSO to TyrA23 p is 0.10, which implies that the reduction may not be statistically significant?

8. Below are a series of related points regarding evidence supporting the incorporation of ApoE into virions, which is insufficient due to lack of proper controls and validation with authentic CCHFV:

Fig. 3C: A control bunyavirus is needed to show that ApoE is present only in CCHFV but not on bunyaviruses that do not use LDLR as an entry factor. For example, HAZV, which the authors show does not depend on LDLR for cellular entry, could be a good control.

Fig. 3F: This is the only experiment shown that tests the effects of apoE incorporation during virus production. More characterization of virions produced in cells that do not express ApoE will be needed. Ideally, Huh-7.5 with apoE knockout or HEK 293T cells overexpressing apoE may be used for controlled comparison. In addition, this panel is missing most statistical analysis. Please show a comparison between anti-LDLR treatment and IgG control for all groups.

Ideally, for a solid claim to be made on apoE incorporation into CCHFV virions, negative stain imaging of the particles should be provided with anti-Gn/Gc gold immunostaining and apoE gold immunostaining.

10. The authors should also provide evidence for lack of Gn/Gc interaction with LDLR, given that the major claim, as noted in the title, is that entry through LDLR is being mediated by apoE displayed on viral particles (which implies that Gn/Gc are not involved).

MINOR:

1. Lines 62-64, "while LDL-R was identified as host entry factor for hepatitis C virus (HCV), hepatitis B virus (HBV), Japanese encephalitis virus (JEV) and vesicular stomatitis virus (VSV)" In the case of VSV, multiple LDLR family members play roles as receptors. Suggest modifying the language here in light of this nuance.

2. Some of the panels are listed with a y-axis noting "% of inhibition compared to 0 ug/ml" and others "% of inhibition compared to no Ab." Would suggest using the latter throughout.

3. Figure 2A: What does the shading represent?

4. Figure 2B: Using "WT" as an abbreviation for "without antibody" is confusing. Suggest using a "No Ab" to avoid confusion with "WT" signifying "wild-type", which it commonly does.

5. Figure 2B: What does "Ctrl" signify? If it is a control, this should be explained in the figure legend

6. Lines 137-138: "This difference between either virus could be due to a different LDL-R usage for the two types of viral particle" Suggest using more precise language here – what is meant by "usage"? Perhaps the authors mean differences in affinity for LDLR?

7. Lines 324-327: "In that context, as both incorporation of human apoE on viral particles and expression of human LDL-R at the surface of human cells enhance infection (our report), LDL-R and apoE might play a crucial role in human-to-human transmission." This statement seems overly general and applicable to most virus entry receptors.

Reviewer #3 (Remarks to the Author):

Ritter et al. provide some in vitro evidence that LDLR plays a role in CCHFV entry events, but some major concerns described below should be addressed to improve the manuscript. Overall, comparisons for statistical analysis to the correct controls are needed in most of the figures, and some standardization is needed with control pseudoviruses, viruses, and VLPs used. Key experiments should also be repeated with wild-type CCHFV to add more convincing data, as it appears to be available to the research group. There appears to be too much reliance on the use of CCHFV VLPs. Given that particle standardization of the VLPs versus pseudovirions and virions is difficult, and experiments utilizing multiple readout modalities between these agents compounds the issue, this is problematic with these types of studies which are quite nuanced. In addition, a more direct method of showing that apoE is incorporated onto/into virions is needed, as the methods used for "purification" only concentrate the virus, along with other debris in the sample, and do not purify it. Once addressed, the findings in this manuscript would be significant in informing medical countermeasure development for CCHFV and contribute greatly to the body of CCHFV research.

Major items:

1. I suggest the use of at least one positive control virus that uses LDLR (such as VSV or SFV) and one negative control virus (that use another known receptor) be included in key in vitro experiments to validate the effects of the knockdowns, pulldowns, and neutralizations and ensure results are not due to non-specific effects or non-specific binding. Some experiments do have at least one of these controls, but there are a few specified below that I suggest these controls also be added. VSV likely also has alternative receptors/entry factors, as it seems CCHFV does, but entry studies should have an abundance of controls in every experiment as they are quite nuanced. It is appreciated that a non-specific controls are included in all experiments, but in the case of virus entry studies these other controls should be considered, particularly when the data is not night and day.
2. The MOI of wild-type virus used for all experiments involving virus should be stated. The method used to standardize the VLPs used should also be stated.
3. Figure 1:
 - a. Figure 1C, 1D, 1G, 1H the statistical analysis for antibody blocking experiments should include the negative control antibodies at each dilution, and not just relative to the no antibody control. For example, in 1D and 1E there appears to be some level of inhibition in both experiments with the IgG goat control and both wild-type CCHFV and wild-type HAZV samples, so the results may not be statistically significant at some concentrations of antibody. For 1D is the level of reduction of CCHFV vRNA significant if you compare the vRNA in the 1 µg/ml of αLDL-R to the same concentration of IgG goat?
 - b. There are no statistic shown in 1E. What is the % infection of HAZV in the controls? What was the MOI of rHAZV-eGFP?
 - c. Figure 1C, for experiments utilizing VLPs or VSVpp with GFP, what was the % positive in the controls? The supplemental data shows the efficiency of the CCHF-VLP with nanoLuc in numerous cell lines, but a statement should be included in the text as to the efficiency of the CCHF-VLP GFP transduction and the VSVpp GFP infection in the controls for these experiments. Given the large difference in the % inhibition between the higher concentrations of αLDL-R and the 0 µg/ml

treated on the graph, it is a wonder that none reached a better p value than * $p < 0.05$, although this is not the correct statistical comparison. The correct comparison for statistics should be the control antibody at the matching concentration, the 0 $\mu\text{g/ml}$ should only be used for normalization. d. Inconsistency of the readouts with controls make comparisons less than ideal. For example, in Fig 1E flow cytometry was used for detection of HAZV-eGFP infection while the NanoLuc detection of CCHF tecVL transduction was used which requires lysing the cells and is a different type of readout (please see minor comment 3 that VLPs do not infect, they are transduced). If wild-type CCHFV could not be used, the use of CCHF tecVLP with a GFP signal and measurement by flow cytometry would be more appropriate for this experiment, as was done for 1C.

e. Figure 1G, a relevant human primary cell would be a good addition to this experiment for biological relevance of utilizing LDL-R for entry, as numerous immortalized (cancer) cell lines upregulate LDL-R (for example A549 - Gueddari et al Biochimie 1993). TE671 appears to have minimal (possibly normal levels) of LDL-R expression, although CCHF-VLPs appear to be readily blocked with α LDL-R antibodies in this cell line as well a relevant primary cell line would be additive.

4. Figure 2:

a. Figure 2A and 2B, binding versus internalization studies with the α -LDLR should be conducted to more definitively show that it would block binding, these may require the use of wild-type virus although qRT-PCR may also work for the VLPs. A better depiction or explanation is needed for the experiment in 2A, what is the "Ctrl" in this experiment? I assume the grey boxes indicate the addition of antibody and duration of incubation with the antibody? Did the H-1 only have the antibody then it was removed and the VLP added, whereas the CTRL had the antibody on through 2 hours post VLP transduction? A 30 minute and 1 hour timepoint should also be included, as by 2 hours it is well past any entry step and there would already be plenty of transcription. The reader has to make too many assumptions regarding the experimental design. The statistics should compare the α LDL- samples to the IgG goat control for each condition and not the WT. The WT should be used for normalization only.

b. The experiment in Figure 2E should also be performed with wild type CCHFV and include a negative control virus (such as HAZV), since this is one experiment that shows a potential direct interaction with CCHFV and LDL-R.

c. Figures 2F and 2G please show negative gating (isotype control) on the flow cytometry histogram. If TyrA23 was left on the cells throughout the 24 hour incubation a greater effect may result, as it appears to in 2H, showing that the uptake of the VLP was dependent on the endocytosis of LDL-R. The addition of a VSV positive control and HAZV negative control would likely strengthen this experiment since the results in 2F are not statistically significant it is not a strong dataset as is.

d. Figure 2H the experiment would be better off combined with the design of 2F, so that there is a direct comparison of no TyrA23 (DMSO only), pre-treatment with TyrA23 only, and pre-treatment and fresh TyrA23. The comparison of the latter to DMSO may be a better comparison and provide significant results.

5. Figure 3:

a. Figure 3A and 3B, a negative control antibody is needed for these experiments, and statistical analysis should be compared to the negative control antibody.

b. Figure 3C pelleting the VLPs in a sucrose cushion is not purifying the virus, the cell debris etc is also pelleted. To purify the VLPs they would need to be harvested from a distinct band in a density gradient.

c. Figure 3D and 3E, a Western blot of enriched VLP showing both viral proteins and ApoE would enhance the data. Doing a pull down of VLP and wild-type virus with Gc and showing ApoE and viral proteins in a Western blot would also enhance this data.

d. Figure 3F, how is the concentration of the VLPs from each cell producer normalized during the addition to the Huh7.5 cells to measure transduction (not infection)? Supplemental Figure 3 indicates that 293T cells may produce either less VLPs, or the lack of ApoE reduces virus uptake thus a lower NanoLuc signal, but this cannot be determined in the results as shown. In addition, the sample sets should not be statistically compared to the no antibody control, they should be compared to the non-specific control antibody. One would suspect that given the low significant value displayed in this figure, the correction to the proper control may eliminate the significance. This experiment would benefit from the use of wild-type virus so that the infectious particles can be quantitated. Line 206-207 should be tempered unless this data is more convincing.

e. Figure 3G, please add information to the legend as to how many technical replicates were run in

this experiment? Was it a single experiment? There are no statistics included, were any of the dilutions statistically significant?

6. Figure 4, suggest adding negative gate (isotype control) to histogram in 4A and 4D.

7. Figure 5, the data as presented also indicate there is an LDL-R independent mechanism of entry so this should be included in the graphical depiction.

Minor items:

1. Use of CCHFV IBAr 10200 should be corrected throughout as "wild-type" and not "full-length" virus throughout, or use CCHFV. As a suggestion CCHFV tecVLPs can also be referred to as CCHF tecVLP or CCHFVLP to avoid confusion with wild-type virus.

2. VLPs do not infect cells as they are not replicating, they are transduced. The use of "infection" regarding VLPs leads to confusion throughout the manuscript as the authors switch back and forth regularly between the use of VLPs, pseudoviruses, and wild-type viruses. Careful editing is needed throughout the manuscript to remove the "infect" or "infection" with regards to VLPs.

3. Figure 1C legend, please add the colors to the legends for CCHF tec VLP and VSVGpp.

4. Figure 1D, Y-axis would be more appropriately labeled as % reduction of vRNA, as it is not a direct readout of % inhibition of infection of wild-type virus.

5. Figure 1F, a more complete description of the samples is needed for the flow cytometry to make it more clear to the reader that the antibody target is on the left of the "/" and the cell type or overexpression type is on the right of the "/". The red VDL-R/WT should state VLDEL-R/WT, correct? Y axis should not state % infection, since this is VLP and the readout is the NanoLuc signal, not % infection of cells.

6. Line 131, Please spell out VSVGpp

7. Line 146 remove typo "ca."

8. Line 206, correct to Figure 3F.

9. Line 211, correct to 3G.

10. Add more details to the materials and methods section:

a. Include assays or references for methods of tiering wild-type or recombinant viruses used, MOIs used in experiments with infectious virus.

b. no methods are included for Figures 3F-H or Figure 4.

c. no methods are included for the flow cytometry of stained cells.

12. Line 297-298 – As written this sentence is confusing. I assume the authors meant that the location of Lrp1 in the membrane may not be proximal to the unknown receptor so there is a possible receptor binding disadvantage because of the distance. The way it is written It sounds like the distance between the virus and cell membrane that may be suboptimal with Lrp1, does this mean there is a large size discrepancy between Lrp1 and LDL-R?

13. Line 328-330, although CCHFV is highly pathogenic it is not necessarily the most pathogenic of all the viruses in the Bunyavirales order, so this statement is unnecessary. I suggest tempering this statement to simply an important pathogenic bunyavirus or something similar.

Reviewer #1 (Remarks to the Author):

Overview: LDLR family members have recently been shown to mediate cellular entry by diverse viruses. Specifically, Lrp1 was shown to be a major host cell entry factor for Rift valley fever virus (RVFV) and Oropouche virus (OROV), both members of Bunyvirales order. Here, Ritter et al present data that the low-density lipoprotein receptor (LDL-R) mediates binding and internalization of Crimean-Congo hemorrhagic fever virus (CCHFV) VLPs and, to a lesser degree, authentic CCHFV, in human hepatocytes but not bovine cells. Interestingly, binding to LDLR does not appear to be due to direct binding by CCHFV GP, but rather binding is mediated by apoE expressed in the viral membrane obtained from the producer cell. This is a new and intriguing mechanism of virus-co-opted entry mechanisms. While the data presented are interesting and exciting, the overall manuscript suffers from data presentation that is confusing and often difficult to interpret, as detailed below. In addition, the majority of the experiments are performed with tecVLPs, where a strong LDLR-dependent phenotype is observed, while only a few experiments are done with authentic CCHFV, and the LDLR-dependent phenotype is less significant. More detailed biochemical (i.e. direct binding between CCHFV GP and LDLR, or lack thereof) and virological (additional studies with wt CCHFV) experiments should be considered to support the authors conclusions. Future in vivo studies to validate these findings should also be considered in the discussion, although it is likely outside the scope of this manuscript. While this manuscript is a fascinating contribution to the field as is, biological significance of these findings would be underscored with additional biochemical, virological, and in vivo experimentation to highlight CCHFV dependence on LDL-R and ApoE.

Reply: we thank this Reviewer for these positive comments that we have thoroughly considered to improve our manuscript.

Major Comments:

1. Explanation of the tecVLP system - what they are, how they are made, and why they are a suitable substitute for authentic virus - needs to be detailed in the manuscript. A diagram or figure may be helpful here. Limitations of this system need to be discussed also.

Reply: we added a panel in Figure 1 to describe the CCHFV tecVLP system used in this study and we give more description of this system in the Introduction (lines 85-88) and in the Methods (lines 509-533) sections. We also discussed the limitations of the system in the Discussion (lines 325-330).

2. Many of the inhibition assays throughout all figures show the data in a counterintuitive manner – i.e. the y-axis is % of inhibition with 0 on the bottom and inhibition rises. Intuitive presentation would be to graph the data as a % of control, starting at 100% and going down.

Reply: we changed all our graphs to express the data as % of infection relative to the corresponding controls.

3. The strong blocking of tecVLPs with anti-LDLR mAb seen in Fig 1C is not replicated in Fig 1D with authentic CCHFV. Same with Fig 2C, 2D, 3A, 3B. For authentic CCHFV studies, additional replicates at each dose should be performed, and actual viral titers should be included for reference. Discussion of the discrepancy between the tecVLP and CCHFV data should be included in the discussion.

Reply: we repeated all the experiments with BSL4 virus to consolidate these data (now included in revised Figures 1e, 2d, 3e). The experiments were performed with different batches of viruses and all experiment was performed with serial dilutions of viruses (corresponding to MOIs of 0.5 to 0.001) at each dose of inhibitors and analyzed with technical replicates. The viral titers of CCHFV stocks (range: 3×10^5 - 10^6 NP FFU/ml) are now indicated in the revised Methods section (lines 495-507). The revised figures convey a clear LDL-R-dependent phenotype for the authentic virus, with levels of inhibition by the anti-LDL-R antibody (Figure 1e) or the soluble LDL-R (Figure 2d) that are similar if not identical to those obtained with the tecVLP assays. The results of the neutralization of the authentic virus with anti-apoE antibody could also be consolidated (revised Figure 3e) though the level of inhibition did not reach those obtained with CCHF tecVLPs. As discussed in the text, this could be due a difference of the number of infectious particles (lines 255-257).

4. Figure 1C and 1D: were there biological replicates in addition to the three independent technical replicates? Are these datapoints an average or only an n=1 from each experiment? This should be clarified if an average of three replicates per experiment replicate are plotted or additional replicates should be performed and those datapoints should be plotted.

Reply: the data presented in Figure 1c and 1d correspond to independent experiments (i.e., biological replicates). For each experiment, serial dilutions of viruses were performed to deduce titers from the linear part of the transduction or infection curves and for each dilution, technical replicates were done.

This is now stated in the revised Methods section (lines 495-507 and 509-533) and in the figure legends.

5. Key experimental details are lacking in the Results section which makes interpretation of the data more difficult.

Examples:

a) Lines 79-81: What method was used to knock down the lipid transfer receptors?

Reply: the knock down assays were performed by transduction of lentiviral vectors encoding for shRNA targeting the different lipid transfer receptors.

We precise this point in the revised Results section (lines 102-105).

b) Lines 84-86: For tecVLP assays here and elsewhere, critical experimental information such as MOI, timepoint harvested, and experimental readout (i.e. luciferase assay vs flow cytometry vs vRNA) need to be included in the results.

Reply: briefly, tecVLPs preparations with titers of ca. 5×10^5 GFP i.u./ml or 10^8 RLU/ml were used in this study. For the different assays used throughout this study, infection assays were performed with serial dilutions of viral stocks, corresponding to MOIs of 2 to 0.02 for the tecVLP-GFP and “RLU-per-cell” of 100 to 0.01 for the tecVLP-NanoLuc. As above-mentioned, the signals obtained with the serial dilutions were used to deduce the actual titers from the linear parts of the transduction or infection curves.

We have added this information in detail in the revised Methods section (lines 509-533) and in less details, in the Results section and the figure legends.

c) Lines 94-96: For CCHFV infection studies, experimental information such as stock/strain used, MOI, time point, readout, and actual titers should be included.

Reply: we have now added this information in details in the revised Methods section, line 495-507, as well as, and in less details, in the Results section and the figure legends.

d) Figs 1C and 2C. Y-axis is % inhibition compared to control, and infectivity (as stated in the legend) is measured by flow cytometry for GFP. This type of experimental details need to be included in the results for proper interpretation of the data. How is GFP expression quantified? Is it % of cells expressing GFP? Or MFI of GFP? It is very unclear in the manner currently presented. Same question for Figs 1D and 2D – how is viral RNA quantified and converted to % inhibition of control?

Reply: we thank this Reviewer to give us the opportunity to clarify this point. We changed all the representations with the Y-axes which are now expressed as “% transduction titers relative to control condition”.

For the assessment of infection by flow cytometry, the % of GFP cells was used to deduce the titers for each experiment by selecting the results for dilutions of virus in the linear range of the signals obtained. We added an example of linearity that we typically obtain in Supplementary Figure 1.

For previous Figures 1D and 2D (now revised Figures 1e and 2d) with WT CCHFV, we infected cells with serial dilutions of viruses (corresponding to MOIs of 0.5 to 0.001). Then, we assessed the levels of viral RNA in infected cell lysates by RT-qPCR and we deduced the infectivity by selecting the virus dilutions in the linear range of detection.

For all the different methods, we then used the titers for each condition to calculate the % of transduction/infection relative to the control conditions.

These different pieces of information in details in the Method section (lines 495-507 and 509-533) but also throughout the Results section and figure legends.

e) Lines 143-147 – how was capture assay performed? Basic information should be included in the Results.

Reply: Both soluble proteins harbor a 6xHis tag and we used Ni-NTA beads to capture these proteins.

We have now added this information line 192 in the Results and line 656 in the Methods sections.

f) Similar to the above comment, in Fig 2F, G, H, data is presented as “level of infection compared to control” (y-axis), and it is difficult to assess whether a change from 1.0 (ctl) to 0.8 (Tyr) is biologically meaningful despite statistical significance. What exactly is being measured (legend indicates GFP expression by flow cytometry) and how is it quantified and compared? If it is % of infection, like Fig 1F, it should be expressed as such. Very difficult to assess the biological relevance of the data on endocytosis in Fig 2 F, G, H.

Reply: we thank this Reviewer to give us the opportunity to clarify this point.

First, we changed the y-axes to “% of transduction titer (fold compared to DMSO)”. Like for the other experiments, we measured the % of GFP positive cells, corresponding to the % of transduction, and then we calculated the titers and compared them in DMSO vs. drug treated conditions.

Second, we repeated these experiments to get stronger statistical significance.

Third, to better explain the data: we showed that TyrA23 treatment could impair LDL-R endocytosis though the effect might not be restricted to this receptor. With a short pre-treatment (30min, Figure 2f), we could detect a relatively mild but reproducible ($p < 0.0001$) effect on tecVLP transduction, concomitant to impairment of LDL-R endocytosis (Supplemental Figure 2c). Yet, with a longer treatment (Figure 2g), the impact of TyrA23 treatment on tecVLP transduction was more difficult to detect since it was confounded by the effect of such a long treatment that induces the accumulation of LDL-R at the cell surface. Thus, in this condition where fresh drug was not replaced at the time of transduction (as in Figure 2g), we could only assess the effect of increased level of LDL-R at the cell surface (and thus probably increased binding) but not only on endocytosis, which promoted rather than inhibited tecVLP transduction. Conversely, when we replenished the pre-treated cells with fresh drug at the time of infection, we detected an impairment of the level of transduction ($p < 0.01$) (Figure 2h), which reflected the effect of impaired endocytosis.

1. Line 173-175 – the word “associate with” or “associate to” is used and the meaning is unclear. Sometimes it seems as if “associate” means “binds” as in Line 173-175 and in other places it may mean something different (lines 97-98).

Reply: we have replaced “associate to” or “associate with” with other terms to clarify our text.

2. Lines 206-207 states that results from Fig 3F indicate that “LDL-R dependent entry of CCHFV does not involve its surface glycoproteins” should be revised because the results indicate that LDLR-mediated CCHFV entry is dependent upon apoE.

Reply: we thank this Reviewer for this point on which we fully agree. We have revised the text accordingly and removed this statement as indeed our evidence that LDLR-mediated CCHFV entry is apoE-dependent does not exclude that LDL-R-dependent CCHFV entry does not involve its surface glycoproteins. Actually, our new results shown in new Figures 3a and 3b indicate that CCHFV GP-pseudotyped VSV particles or tecVLPs are still dependent on LDL-R when they are produced in apoE-deficient 293T cells (though much less than when tecVLPs are produced in apoE-expressing Huh-7.5 cells), which corroborates the findings of other (Xu *et al.*, Cell Res., 2024) that LDL-R dependent entry of CCHFV involves its surface glycoproteins. Nevertheless, our results indicate that apoE is recruited by CCHFV particles to induce LDLR- entry.

3. Fig 4 would be bolstered by inclusion of fluorescence microscopy to visualize downregulation of surface expression of LDLR after treatment with PCSK9.

Reply: As proposed by this Reviewer, we performed cell surface staining of LDL-R on cells treated with or without PCSK9 (as well as with or without Berbamine (BBM) treatment) and imaged them by confocal microscopy. As a control, we stained CD81 proteins as it is also expressed at the cell surface. We found that while BBM or PCSK9 treatment did not change CD81 signal at the cell surface, these treatments drastically impaired LDL-R levels at the cell surface.

We have added these data as panels B and F in the revised Figure 5.

Minor comments:

1. Line 78: why were Huh 7.5 cells chosen for use and what LDLRs do they naturally express?

Reply: we selected Huh7.5 cells as they are in our hand the human cell that are the most permissive to both CCHFV infection and tecVLP production/transduction, as tested by us among a panel of 20 other cell types. Based on the literature and our own experience, we know that they express LDL-R, Lrp1 as well as SR-B1 but that they do not express VLDL-R in normoxic conditions.

We have added this information in the revised text (lines 98-101).

2. Fig 1B: y-axis is “Level of Infection (compared to control cells)” is unclear. Is this % infection?

Reply: this Reviewer is right, the y-axis should be “% of infection compared to respective control condition”. We corrected the y-axis of this Figure (now revised Figure 1c) to “% of transduction titers compared to control cells”.

3. Fig 1F: the orange/red/blue text legend on the left is unclear as to what it is referring to.

Reply: the color code corresponds to the conditions of the flow cytometry staining. We clarified the legend.

4. Fig 2B: graph is presented horizontally instead of vertically which makes it difficult to interpret compared to similar graphs throughout the manuscript. Suggest keeping the orientation consistent throughout.

Reply: we changed this panel for a vertical graph for better clarity.

5. Fig 3A,B: x-axis should indicate concentration of anti-apoE, rather than dilution

Reply: we used an anti-apoE serum for which we do not have the concentration, explaining why we expressed data with dilutions and not concentrations.

6. Fig 3G: This should not be a line graph and instead should be a bar graph with data points as in Fig 3A.

Reply: we corrected this graph which is now new Figure 3f.

7. Line 86/243: It appears the shRNA downregulation of Lrp1 promoted infection of Huh7.5 cells. LDL-R family receptors often compensate for one another. Therefore the authors should test if shRNA KD of Lrp1 resulted in an increase in LDL-R surface expression which could then explain the increase in infectivity observed. Alternatively, KD of both or multiple LDL-R family receptors.

Reply: we thank the Reviewer for this question. We checked both the total level of LDL-R and the level of LDL-R at cell surface upon Lrp1 KD. Interestingly, we found that while the total level of LDL-R was not changed, the level at cell surface was increased, suggesting that Lrp1 KD might promote CCHFV tecVLPs transduction by upregulating LDL-R at cell surface, although we could not exclude an antiviral effect of Lrp1.

We added the data as supplementary Figures 1a and 1b and in the text (lines 110-113).

8. Line 99: since other viruses utilize LDL-R, it is not specifically only used by CCHFV. It may be more accurate to state “suggesting that LDL-R is not a pan-orthonairovirus entry factor”

Reply: this Reviewer is right and we corrected the statement (lines 136-137)

9. Line 136: Similar to the use of HAZV in Figure 1, a negative control virus should be used in the sLDL-R neutralization assay (Figure 2C) to demonstrate that it does not have a pan anti-viral impact on Huh7.5 cells

Reply: we thank this Reviewer for this comment. We repeated the sLDLR neutralization with MLV pseudoparticles (MLVpp) and with HAZV, which were both insensitive to sLDL-R neutralization. The data are included in new Figure 2c.

10. In general, there is no comment on biosafety level of CCHFV, biological containment, and inactivation of samples. This reviewer believes this would add to the significance of investigating this virus. This is usually mentioned in the methods. In addition, no details are given on the q-RT-PCR assay used for CCHFV.

Reply: we added the information on CCHFV manipulation in the Introduction, Results, Discussion and in Methods sections. We also provide details on the RT-qPCR assay used to determine CCHFV infectivity.

11. Line 206: Figure 4F should be Figure 3F

Reply: we thank this Reviewer for pointing this out and we corrected the text.

12. Line 211: Figure 4G should be Figure 3G

Reply: we thank this Reviewer and we corrected the text.

13. Line 226: Similar to the use of HAZV in Figure 1, another virus should be shown as a negative control to demonstrate that the slight decrease in viability (although not significant) following PCSK9 treatment does not impact general viral infection in a non-LDL-R manner

Reply: we thank this Reviewer for this point. We repeated PCSK9 treatment with control viruses including HAZV and lentiviral vectors pseudotyped with MLV-A Env glycoprotein (MLVpp) or VSV-G (VSVpp). We found that PCSK9 treatment only affected VSVpp and CCHFV tecVLP transduction without impairing MVLpp transduction and HAZV infection.

We have included these data as Figure 5d (and Figure 5h for BBM) and we modified the text accordingly (lines 306 and 311).

Editorial points to help with readability:

- Line 28: “cause” should read “can cause”
- Line 31: “as CCHFV” should read “as a CCHFV entry factor”
- Line 31: remove “could” and instead write “LDL-R promoted binding and internalization”
o This line in general is confusing: “LDL-R could promote binding and internalization of CCHFB particles” perhaps instead it should read “LDL-R facilitated binding and internalization...”
- Line 33: “mediate” should be “mediates the LDL/LDL-R”
- Line 34: “incorporated on CCHFV” should perhaps be “incorporated in CCHFV particles”
- Line 36: “hijacked” could instead be “utilizes” or “takes advantage of”
- Line 44: “i.e.” should be removed
- Line 89: delete “of” and “by”
- Line 92: either define KD in line 79 or spell out
- Line 102: delete “the”. It should read “with the exception of hepatocytes”
- Line 146: “ca.”? what does this mean. Perhaps spell out “approximately”?
- Line 151: should read “we determined the effect of TyrA23 on both” or “TyrA23’s”
- Line 155: should read “impairs” instead of “impaired”
- Line 180: should read “As a positive control, ...”
- Line 181: should read “as a negative control..”
- Line 203: should read “(Supplemental Figure 3A) but are fully able to...”
- Line 219: decreases” should read “decreased”
- Line 220: readability would improve if it read “LDL-R expression at the cell surface”
- Line 224: “level” should read “levels”

- Line 261: should read “combined with our report”
- Line 267/268: delete “e.g.”
- Line 274: should read “bind to a large variety of ligands”
- Line 284: “factor” instead of “factors”
- Line 288: citation formatting needs to be edited
- Line 298: missing a . between “membranes” and “In”
- Line 299: “raise” should be “raises”
- Line 323: underline should be removed
- Line 361: “R&D System” should read “systems”
- Line 364/365: anti should possibly be capitalized
- Line 386: there should be an “and” before “filtered”
- Line 416: “filtered” spelling error
- Line 421: “Firefly Luciferase cells”? do the authors mean “Huh7.5 cells expressing firefly luciferase were transduced..”
- Line 452: do authors mean “Inocultes were incubated”?
- Line 727 in Figure 1 legend: “stably expressing Fluc were transduced” (“cells” should be deleted)
- Line 746 in Figure 2 legend: “ON” should be clarified as “overnight”
- Line 763: “were treated with DMSO” (“for” should be deleted)
- Line 810/811 etc: symbols

Reply: we thank very much this Reviewer for these editorial points that we corrected throughout the text.

Reviewer #2 (Remarks to the Author):

Ritter et al. report on the identification of low-density lipoprotein receptor as a cellular receptor for the tick-borne emerging pathogen Crimean-Congo hemorrhagic fever virus (CCHFV). They used a series of approaches including antibody blocking studies and blocking studies with soluble receptor fragments. They further attempted to show that the LDLR ligand, apoE, was incorporated into viral particles and was responsible for LDLR-mediated entry, although the latter was only demonstrated using some indirect assays (e.g., showing that tecVLPs have apoE when they are produced in huh7 cells, that antibody against apoE can block entry of tecVLPs and authentic virus, and that anti-apoE antibody can enrich CCHFV minigenome RNA in a pull-down assay). Overall, while the work implicating LDLR as a cellular entry factor is compelling, the findings implicating the role of ApoE as an intermediate factor that is required for LDLR engagement are less robust. Critical controls are lacking, and experiments exploring a direct interaction between the Gn/Gc envelope proteins with LDLR are also missing. Lastly, the data with fully infectious virus have a great deal of variability which makes the robustness of the findings hard to appreciate. Additional concerns in regards to the statistical analysis are listed below.

Reply: we thank this Reviewer for his/her important comments that we have all taken into account by performing additional and/or new experiments.

MAJOR:

1. Figure 1D is a critical experiment attempting to confirm the results of antibody-blocking studies performed with surrogate systems (e.g., tecVLPs) but with authentic virus. The variability between the data points is quite high (from 0 to 70%). Could the authors comment on the likely source of variability?

Reply: we thank this Reviewer for this point. For authentic virus, all the experiments were done in BSL-4 which induces experimental limitations. The variability can be explained by different factors. First, we measured the level of infection by detecting viral RNA by qPCR in infected cell lysates, which is highly sensitive but according to the qPCR rule, “3.3 Ct roughly equals 1log molecules”, cannot be compared to the accuracy of a flow cytometry reading that was used in most experiments with tecVLPs.

Second, experiments with either tecVLPs or WT virus were done with independent batches of viruses. While the production of tecVLPs relies on plasmid transfection and is quite reproducible, the production of WT CCHFV relies on infection and amplification of cells with a non-clonal virus stock, which is likely to induce more variations from batch to batch owing virus propagation that is difficult to control between experiments.

Nevertheless, we repeated the experiments in BSL-4 in order to increase the reproducibility of the results. The new data set is presented in revised Figure 1e (and in revised Figures 2d, 3e and 4d).

2. Figure 2D, which includes soluble LDLR blocking assays with authentic virus – again, like with the antibody blocking studies, the variability in these experiments is quite high. I have the same points to raise as those noted above.

Reply: as Figure 2d is based on the same read-out and same batches of WT virus than revised Figure 1e, the explanations for the variability in these experiments are the same than described above.

Like for revised Figure 1e, we repeated the experiments, which reduced variability.

3. Figure 2E. Is the capture of viral RNA by CD81-LEL nonspecific? It seems like a “beads alone” control is missing in this experiment (e.g., in this case, based on the methods, this would be “Ni-particles” alone).

Reply: we compared CD81-LEL binding to beads alone and it seems that the capture by CD81-LEL corresponds to a nonspecific bead binding as the signal is the same than capture with beads alone (Figure below).

4. Lines 104-105: “Interestingly, expression of VLDL-R increased infection levels of tecVLPs by 2-3 folds (Figure 1F)” It looks like the change may not be statistically significant ($p=0.0625$) - this should be noted also in the main text.

Reply: we repeated this experiment and, while the high p value ($p=0.0625$) in the previous figure is likely due to an outlier point (500%), we now confirm that the change of infection levels upon over-expression of VLDL-R is statistically significant ($p<0.05$, revised Figure 1f).

5. Statistical analyses should be made between control and treatment, not just across concentration gradients. In Fig. 1C-E, 2B, 3A-B, 3F: please show comparison between anti-LDLR treatment and control IgG.

Reply: we thank this Reviewer for this comment. We have modified our statistical analyses and we now provide all the comparisons between control and treatment in the revised figures.

6. Fig. 3A and 3B - Control IgG should be included in these experiments.

Reply: we have included new data in the revised version of this experiment (now Figures 3d and 3e), including results of control goat serum.

Note that we have also repeated this experiment with additional control viruses (i.e., MLVpp and HAZV). The related text is in lines 255-259.

7. Lines 166-168: “This treatment prevented both LDL-R endocytosis (Supplemental Figure 2D) and tecVLP infection (Figure 2H), suggesting that LDL-R is involved in CCHFV endocytosis.” The p value in Figure 2H comparing DMSO to TyrA23 p is 0.10, which implies that the reduction may not be statistically significant?

Reply: we have repeated this TyrA23 experiment, and we now show in revised Figures 3f-3h statistically significant differences.

8. Below are a series of related points regarding evidence supporting the incorporation of ApoE into virions, which is insufficient due to lack of proper controls and validation with authentic CCHFV:

Reply: we have considerably reinforced the evidence supporting the involvement of ApoE into CCHFV entry, through its incorporation into virions. Accordingly, the initial figure dedicated to this important point (Figure 3) has been doubled into Figure 3 and Figure 4 to display all the revised and new results. The experiments addressing the specific point on apoE incorporation into virions are summarized below:

- Figure 3a (new): LDL-R dependency assays with VSV particles pseudotyped with VSV-G, CCHFV GPC and EBOV GP.
- Figure 3b (revised/new): LDL-R dependency assays with tecVLPs produced in Huh-7.5 vs. HEK293T cells using either Huh-7.5 or HEK293T as target cells.

- Figure 3c (new): LDL-R dependency assays with tecVLPs produced in HEK293T cells vs. in HEK293T cells that ectopically express apoE.
- Figure 4a (revised): detection of apoE in pelleted tecVLPs.
- Figure 4b (new): detection of apoE in pelleted HAZV particles.
- Figure 4c (revised): co-capture of tecVLP RNAs by apoE immuno-precipitation, using HAZV as control particles.
- Figure 4d: Gn or apoE immunogold labeling of tecVLPs analyzed by electron microscopy.
- Figure 4e (new): co-capture of WT CCHFV RNAs by apoE immuno-precipitation.
- Figure 4f (new): co-capture of WT CCHFV Gn and Gc GPs by apoE immuno-precipitation.
- Figure 4h to 4j (new): production and infectivity of tecVLPs from Huh-7.5 cells vs. apoE KD Huh-7.5 cells, using HAZV as control particles.

To specifically address the points of this Reviewer:

Fig. 3C: A control bunyavirus is needed to show that ApoE is present only in CCHFV but not on bunyaviruses that do not use LDLR as an entry factor. For example, HAZV, which the authors show does not depend on LDLR for cellular entry, could be a good control.

Reply: we thank the Reviewer for this point. We repeated the same process of purification of viral particles with ultracentrifugation with HAZV. Interestingly, we could not observe any increase of apoE in pellet containing HAZV virions. We added the data as new data Figure 4b.

Fig. 3F: This is the only experiment shown that tests the effects of apoE incorporation during virus production. More characterization of virions produced in cells that do not express ApoE will be needed. Ideally, Huh-7.5 with apoE knockout or HEK 293T cells overexpressing apoE may be used for controlled comparison. In addition, this panel is missing most statistical analysis. Please show a comparison between anti-LDLR treatment and IgG control for all groups.

Reply: we thank this Reviewer for this comment. We considerably expanded our results on apoE in our revised manuscript.

First, we consolidated the data of Figure 3F (now Figure 3b) by adding new repeats but also by performing new experiments with HEK293T as target cells in addition to Huh-7.5 cells. The statistical analyses of the results of anti-LDLR treatment and IgG control for all conditions are now significant. Second, as proposed by this Reviewer, we produced CCHFV tecVLPs in Huh-7.5 cells in which apoE was knock-down by shRNA. Strikingly, we could observe a strong loss of transduction of the resulting particles, which resulted from both lower secretion and lower specific infectivity of viral particles. While apoE KD had no effect on HAZV production used a control in these experiments, these results indicate that apoE is actively recruited on CCHFV particles and is involved in assembly/secretion, which is reminiscent of the interrelation between apoE and HCV particles, as put forward in the revised Discussion. We included this data set in new Figure 4 (panels 4g to 4j).

Third, we overexpressed apoE in HEK293T cells that produce tecVLPs and we found that this promoted the transduction titer and the sensitivity of these tecVLPs to LDL-R blocking. We added these data as new Figure 3c and Supplementary Figure 3g.

The related text reads in lines 228-245 and 286-295.

Ideally, for a solid claim to be made on apoE incorporation into CCHFV virions, negative stain imaging of the particles should be provided with anti-Gn/Gc gold immunostaining and apoE gold immunostaining.

Reply: as requested by this Reviewer, thanks to a collaboration with Pr. Philippe Roingard, we performed a negative stain imaging of tecVLPs combined with either anti-Gn or anti apoE gold immunostaining. The amounts of detected particles were too low to combine both anti-Gn and anti-apoE staining, though we could detect viral particles with either anti-Gn or anti-apoE staining.

We added these data as new Figure 4d and the related text (lines 281-282).

10. The authors should also provide evidence for lack of Gn/Gc interaction with LDLR, given that the major claim, as noted in the title, is that entry through LDLR is being mediated by apoE displayed on viral particles (which implies that Gn/Gc are not involved).

Reply: we thank the Reviewer to raise this important point. To address this question, we produced VSV-ΔG particles pseudotyped with VSV-G, which directly binds LDL-R, or with CCHFV GPs (or with EBOV GP, as control) in HEK293T cells that do not express apoE. When we tested the effect of LDL-R blocking on infection with these particles, we found that while this strongly reduced infection of VSV-G pseudotypes, as expected, it had only a mild effect on CCHFV GP pseudotypes (and no effect on EBOV GP pseudotypes), which compared with the strong impact of this blocking on CCHFV tecVLPs or on WT virus that are produced in Huh-7.5 cells (apoE expressing). Furthermore, when we compared such blocking assays with CCHFV tecVLPs produced in Huh-7.5 vs. HEK293T cells, we found that the former were more sensitive to this blocking than the latter. These two results, now displayed in new Figure 3a and Figure 3b and highlighted in the revised text (lines 223-238), led us to deduce that while CCHFV GPs allow some levels of interaction with LDL-R, apoE can dramatically sustain the interaction.

In the meantime, an article showed an interaction between Gc and LDL-R was published during the revision of our paper (Xu *et al.*, 2024, Cell Res). Thus, based on this publication and of our new data, we modulated our statement to reflect the evidence that apoE can promote CCHFV entry via LDL-R. We also changed the title of our manuscript.

MINOR:

1. Lines 62–64, “while LDL-R was identified as host entry factor for hepatitis C virus (HCV), hepatitis B virus (HBV), Japanese encephalitis virus (JEV) and vesicular stomatitis virus (VSV)”. In the case of VSV, multiple LDLR family members play roles as receptors. Suggest modifying the language here in light of this nuance.

Reply: we have changed this sentence (lines 80-82) to reflect that multiple LDLR family members play roles as receptors for VSV.

2. Some of the panels are listed with a y-axis noting “% of inhibition compared to 0 ug/ml” and others “% of inhibition compared to no Ab.” Would suggest using the latter throughout.

Reply: we thank the Reviewer for this point that has now been corrected throughout the text. Note that our results are now expressed as “% infection/transduction” rather than “% inhibition” to comply with the request of Reviewer #1 (point #2).

3. Figure 2A: What does the shading represent?

Reply: we corrected Figure 2a to improve clarity. The shading corresponds to the time when the antibodies are present. We added “+ Ab” in the figure and indicated this in the figure legend.

4. Figure 2B: Using “WT” as an abbreviation for “without antibody” is confusing. Suggest using a “No Ab” to avoid confusion with “WT” signifying “wild-type”, which it commonly does.

Reply: we corrected this point.

5. Figure 2B: What does “Ctrl” signify? If it is a control, this should be explained in the figure legend

Reply: this “Ctrl” condition corresponded to the protocol used in Figure 1. To clarify this, we changed the labelling of this condition to “H-1 & H0”, i.e. meaning that the antibodies were present before and during the transduction.

6. Lines 137-138: “This difference between either virus could be due to a different LDL-R usage for the two types of viral particle” Suggest using more precise language here – what is meant by “usage”? Perhaps the authors mean differences in affinity for LDLR?

Reply: we corrected the term “usage” by “role”. Our point was to hypothesize that LDL-R might play different role in entry of these particles; yet, this Reviewer is right that affinity could also explain the discrepancy. We also added this point (lines 185-186).

7. Lines 324-327: “In that context, as both incorporation of human apoE on viral particles and

expression of human LDL-R at the surface of human cells enhance infection (our report), LDL-R and apoE might play a crucial role in human-to-human transmission.” This statement seems overly general and applicable to most virus entry receptors.

Reply: we have modified this sentence to “In that context, as incorporation of apoE on viral particles enhances infection through LDL-R (our report), apoE might play a crucial role in tropism and/or reservoir establishment in a manner dependent on tissue-specificity of apoE expression, such as in hepatocytes.” (lines 444-447).

Reviewer #3 (Remarks to the Author):

Ritter et al. provide some in vitro evidence that LDLR plays a role in CCHFV entry events, but some major concerns described below should be addressed to improve the manuscript. Overall, comparisons for statistical analysis to the correct controls are needed in most of the figures, and some standardization is needed with control pseudoviruses, viruses, and VLPs used. Key experiments should also be repeated with wild-type CCHFV to add more convincing data, as it appears to be available to the research group. There appears to be too much reliance on the use of CCHFV VLPs. Given that particle standardization of the VLPs versus pseudovirions and virions is difficult, and experiments utilizing multiple readout modalities between these agents compounds the issue, this is problematic with these types of studies which are quite nuanced. In addition, a more direct method of showing that apoE is incorporated onto/into virions is needed, as the methods used for “purification” only concentrate the virus, along with other debris in the sample, and do not purify it. Once addressed, the findings in this manuscript would be significant in informing medical countermeasure development for CCHFV and contribute greatly to the body of CCHFV research.

Reply: we thank this Reviewer for his/her thoughtful commentaries that we have fully considered to improve our manuscript.

Major items:

1. I suggest the use of at least one positive control virus that uses LDLR (such as VSV or SFV) and one negative control virus (that use another known receptor) be included in key in vitro experiments to validate the effects of the knockdowns, pulldowns, and neutralizations and ensure results are not due to non-specific effects or non-specific binding. Some experiments do have at least one of these controls, but there are a few specified below that I suggest these controls also be added. VSV likely also has alternative receptors/entry factors, as it seems CCHFV does, but entry studies should have an abundance of controls in every experiment as they are quite nuanced. It is appreciated that a non-specific controls are included in all experiments, but in the case of virus entry studies these other controls should be considered, particularly when the data is not night and day.

Reply: We thank this Reviewer for these encouraging comments as indeed, these data are not easy to acquire. We paid a great attention to find the most suitable, sometimes not so easy control viruses. Accordingly, as requested by this Reviewer, we have repeated and added controls in the key experiments, e.g. in revised Figures 1d, 2c, 2e, 3d, 4c, 5d and 5h, and in new Figures 3a, 4b, 4g-4j. We believe that the overall message is now more solid and convincing.

2. The MOI of wild-type virus used for all experiments involving virus should be stated. The method used to standardize the VLPs used should also be stated.

Reply: we thank this Reviewer to raise this point. We have stated in the revised Methods section (line 502-507) the MOIs of the wild-type virus used in the experiments.

Regarding the method used to standardize the VLPs, as stated in the revised Methods section (lines 509-533), preparations of tecVLPs with titers of ca. 5×10^5 GFP i.u./ml (for tecVLP-GFP) or 10^8 RLU/ml (for tecVLP-NanoLuc) were used in the experiments. We always used serial dilutions of tecVLPs (or other viral particle types), as stated in the text and in the figure legends, to ascertain that we could deduce titers from the linear range of the signals obtained experimentally. This was particularly critical for tecVLPs harboring a GFP minigenome since the levels of transduction (i.e., % of GFP-positive cells) are not only dependent on the inoculum size but also on the efficiency of NP+L pre-transfection of target cells (which allows minigenome replication and GFP amplification).

3. Figure 1:

a. Figure 1C, 1D, 1G, 1H the statistical analysis for antibody blocking experiments should include the negative control antibodies at each dilution, and not just relative to the no antibody control. For example, in 1D and 1E there appears to be some level of inhibition in both experiments with the IgG goat control and both wild-type CCHFV and wild-type HAZV samples, so the results may not be statistically significant at some concentrations of antibody.

For 1D is the level of reduction of CCHFV vRNA significant if you compare the vRNA in the 1 $\mu\text{g/ml}$ of $\alpha\text{LDL-R}$ to the same concentration of IgG goat?

Reply: We repeated the experiments and we revised the statistical analyses to compare $\alpha\text{LDL-R}$ and ctrl IgG for each dilution. All results in the corresponding revised Figures 1d, 1e, 1f, 1i and 1j are now statistically significant.

b. There are no statistic shown in 1E. What is the % infection of HAZV in the controls? What was the MOI of rHAZV-eGFP?

Reply: as explained in point 2 and in the revised Method section (lines 535-545), we applied serial dilutions of the virus inoculate for the experiments. These dilutions corresponded to MOIs ranging from 0.5 to 0.001. The linear range was achieved for % between 5% to 20%. We added a representative example of these assays as Supplementary Figure 1c. As for previous Figure 1e, the experiment has been repeated and we added another control (MLVpp): this is now included in revised Figure 1c with the requested statistical analysis.

c. Figure 1C, for experiments utilizing VLPs or VSVpp with GFP, what was the % positive in the controls? The supplemental data shows the efficiency of the CCHF-VLP with nanoLuc in numerous cell lines, but a statement should be included in the text as to the efficiency of the CCHF-VLP GFP transduction and the VSVpp GFP infection in the controls for these experiments. Given the large difference in the % inhibition between the higher concentrations of $\alpha\text{LDL-R}$ and the 0 $\mu\text{g/ml}$ treated on the graph, it is a wonder that none reached a better p value than * $p < 0.05$, although this is not the correct statistical comparison. The correct comparison for statistics should be the control antibody at the matching concentration, the 0 $\mu\text{g/ml}$ should only be used for normalization.

Reply: we thank this Reviewer for highlighting these points that we have addressed in detail in the revised Methods section.

We performed the experiments in Figure 1C and others by applying serial dilutions of viruses/VLPs, which could be different depending on the viral set up used, and we deduced their titers from selected values within the linear range of detection of positive cells. This is now indicated in the revised text and figure legends. We also added in Supplementary Figure 1c a representative example of percentage of GFP positive cells as a function of the viruses/VLPs inoculates.

We repeated these experiments of previous Figure 1c (now revised Figure 1d), we added control viruses, and we corrected the statistical analyses by comparing $\alpha\text{LDL-R}$ with control Ig for each condition. The results now display a high level of confidence.

d. Inconsistency of the readouts with controls make comparisons less than ideal. For example, in Fig 1E flow cytometry was used for detection of HAZV-eGFP infection while the NanoLuc detection of CCHF tecVL transduction was used which requires lysing the cells and is a different type of readout (please see minor comment 3 that VLPs do not infect, they are transduced). If wild-type CCHFV could not be used, the use of CCHF tecVLP with a GFP signal and measurement by flow cytometry would be more appropriate for this experiment, as was done for 1C.

Reply: we thank this Reviewer to raise this important point.

To explain why we performed these experiments in this way in the previous Figure 1E: we were aware that the different readouts are a concern and we internally discussed this question in depth before deciding to perform these experiments that way. Specifically for this experiment, we needed to use tecVLPs with nanoLuc minigenome since, for detection of transduction of tecVLPs with GFP minigenome, target cells need to be pre-transfected for expression of CCHFV L and NP. However, this would have required to also use such CCHFV L+NP pre-transfected cells while testing HAZV since the experiments needed to be done side-by-side. Yet, we preferred not to do so for safety reasons, i.e., to exclude the possibility of formation of a chimeric HAZV particles. Thus, since pre-transfection is not required for handling tecVLPs with a nanoLuc minigenome, this allowed us to use exactly the same target cells for either viral set up.

Nevertheless, to comply with the request of this Reviewer, we repeated the experiments using tecVLPs with a GFP minigenome (using L+NP pre-transfected target cells) and with HAZV, VSVpp and MLVpp as controls. These data are now presented as new Figure 1d. As for the LDL-R blocking assay with wild-type CCHFV (revised Figure 1e), the readout is different than for HAZV-eGFP; thus, we preferred to display the former in a separate panel.

e. Figure 1G, a relevant human primary cell would be a good addition to this experiment for biological relevance of utilizing LDL-R for entry, as numerous immortalized (cancer) cell lines upregulate LDL-R (for example A549 - Gueddari et al Biochimie 1993). TE671 appears to have minimal (possibly normal levels) of LDL-R expression, although CCHF-VLPs appear to be readily blocked with α LDL-R antibodies in this cell line as well a relevant primary cell line would be additive.

Reply: as requested by this Reviewer, we performed blocking experiments with α LDL-R antibody in primary human hepatocytes (PHH), as the liver and hepatocytes are relevant CCHFV targets. We could confirm that CCHFV tecVLP transduction of PHH is impaired by α LDL-R antibody.

We added the data as new Figure 1h and modified the text accordingly (lines 146-149). We also provide in new Figure 1g the LDL-R surface staining of the different cell types studied in these blocking assays, including the PHH.

4. Figure 2:

a. Figure 2A and 2B, binding versus internalization studies with the α -LDLR should be conducted to more definitively show that it would block binding, these may require the use of wild-type virus although qRT-PCR may also work for the VLPs. A better depiction or explanation is needed for the experiment in 2A, what is the "Ctrl" in this experiment? I assume the grey boxes indicate the addition of antibody and duration of incubation with the antibody? Did the H-1 only have the antibody then it was removed and the VLP added, whereas the CTRL had the antibody on through 2 hours post VLP transduction? A 30 minute and 1 hour timepoint should also be included, as by 2 hours it is well past any entry step and there would already be plenty of transcription. The reader has to make too many assumptions regarding the experimental design. The statistics should compare the α LDL- samples to the IgG goat control for each condition and not the WT. The WT should be used for normalization only.

Reply: we apologize for our lack of clarity when depicting this experiment. In Figures 2A and 2B, our aim was to discriminate LDL-R involvement in entry (i.e. binding + internalization + fusion) vs. transcription/replication steps, hence the selected time points for this first experiment that address his/her specific questions. This Reviewer is right to say that the grey boxes indicate when the antibody is present and we thus clarified the experimental design in the revised Figure 2a.

The Ctrl corresponds to conditions used in Figure 1, i.e. blocking with anti-LDL-R antibody for 1h before infection followed by infection in presence of fresh antibody. Accordingly, we changed the misleading "ctrl" term by "H-1 & H0".

As requested, we also corrected the statistical analyses by comparing for each condition anti-LDLR vs. control IgG.

b. The experiment in Figure 2E should also be performed with wild type CCHFV and include a negative control virus (such as HAZV), since this is one experiment that shows a potential direct interaction with CCHFV and LDL-R.

Reply: As requested by this Reviewer, we performed sLDL-R capture experiments with WT CCHFV and HAZV. The results displayed in revised Figure 2e (right graphs) and described in the text (lines 194-196) show that WT CCHFV but not HAZV can be captured by sLDL-R.

c. Figures 2F and 2G please show negative gating (isotype control) on the flow cytometry histogram. If TyrA23 was left on the cells throughout the 24 hour incubation a greater effect may result, as it appears to in 2H, showing that the uptake of the VLP was dependent on the endocytosis of LDL-R. The addition of a VSV positive control and HAZV negative control

would likely strengthen this experiment since the results in 2F are not statistically significant it is not a strong dataset as is.

Reply: We thank the Reviewer for these comments, which allowed us to improve these figures set.

First, we added the isotype controls on the flow cytometry histograms of Figures 2f and 2g. Second, we repeated the experiments with CCHFV tecVLPs and achieved statistically significant differences (revised Figures 2f, 2g and 2h).

Third, as requested by this Reviewer, we also tested TyrA23 on HAZV and VSVpp. Regarding HAZV, we could see an inhibition of HAZV infection upon treatment (Figure A, below); yet, as TyrA23 is not specific to LDL-R, it may inhibit an HAZV entry through its proper endocytosis pathway that remains not fully defined for this virus. Regarding VSVpp, we did not observe an effect of TyrA23 treatment. The likely explanation for this apparent paradox is a different entry pathway for either CCHFV or VSV. Indeed, while LDL-R endocytosis and CCHFV tecVLPs entry are both sensitive to chlorpromazine (Supplementary Figure 2), this is not the case for VSVpp entry (Figure B, below), suggesting that in our cells, VSVpp and LDL-R do not follow the same endocytosis pathway and hence, that VSVpp might not be an appropriate positive control.

Because these results suggest that they are not appropriate controls, we would prefer to not include them in the revised text.

d. Figure 2H the experiment would be better off combined with the design of 2F, so that there is a direct comparison of no TyrA23 (DMSO only), pre-treatment with TyrA23 only, and pre-treatment and fresh TyrA23. The comparison of the latter to DMSO may be a better comparison and provide significant results.

Reply: we thank this Reviewer for this comment and we represented the data as requested. Of note, there is no difference between DMSO only vs. pre-treatment + fresh TyrA23, since the latter condition has opposite effects: while pre-treatment increased LDL-R presence at the cell surface and thus increased CCHFV tecVLPs transduction, the addition of TyrA23 at the time of transduction decreased the infection by blocking endocytosis.

5. Figure 3:

a. Figure 3A and 3B, a negative control antibody is needed for these experiments, and statistical analysis should be compared to the negative control antibody.

Reply: we added negative control antibody results of this experiment, as requested, and we performed the statistical analysis by comparison to these control conditions. These data correspond to new Figures 3d and 3e, that also include additional control viruses (HAZV and MLVpp).

b. Figure 3C pelleting the VLPs in a sucrose cushion is not purifying the virus, the cell debris etc is also pelleted. To purify the VLPs they would need to be harvested from a distinct band in a density gradient.

Reply: we agree with this Reviewer that pelleting the VLPs in a sucrose cushion is not the best method to purify the virus, though we did not detect actin from putative cell debris in these pellets, as shown in this Figure (now revised Figure 4a).

To improve the demonstration that apoE is associated to CCHFV particles, first, we repeated the tecVLP RNA pull down experiments with apoE antibodies, which now shows statistically significant results as displayed in revised Figure 4c. Second, we performed pull down assays with WT CCHFV particles and detected strong enrichment of viral RNA and of both Gn and Gc proteins. These results are shown in the revised Figures 4e and 4f, and in the text lines 277-285.

c. Figure 3D and 3E, a Western blot of enriched VLP showing both viral proteins and ApoE would enhance the data. Doing a pull down of VLP and wild-type virus with Gc and showing ApoE and viral proteins in a Western blot would also enhance this data.

Reply: see our above reply (point b). We believed that these new results with WT CCHFV strongly enhance our statement that apoE associates with CCHFV particles.

d. Figure 3F, how is the concentration of the VLPs from each cell producer normalized during the addition to the Huh7.5 cells to measure transduction (not infection)? Supplemental Figure 3 indicates that 293T cells may produce either less VLPs, or the lack of ApoE reduces virus uptake thus a lower NanoLuc signal, but this cannot be determined in the results as shown. In addition, the sample sets should not be statistically compared to the no antibody control, they should be compared to the non-specific control antibody. One would suspect that given the low significant value displayed in this figure, the correction to the proper control may eliminate the significance. This experiment would benefit from the use of wild-type virus so that the infectious particles can be quantitated.

Reply: we thank this Reviewer for this opportunity to clarify our experimental set up. The production of tecVLPs in HEK293T cells raises ca. 5-fold lower titers compared to production in Huh-7.5 cells; thus, to compare the two preparations, we adapted the serial dilutions in order to achieve comparable % of GFP positive cells or nanoLuc signals. We added this information as Supplementary Figure 3d-3e.

We repeated the experiments and compared Huh-7.5 and HEK293T as target cells and added the requested statistical analysis. This is now depicted as new Figure 3b and in the text, lines 228-238.

Line 206-207 should be tempered unless this data is more convincing.

Reply: we agree with this Reviewer. Indeed, the revised and new data displayed in Figures 3a and 3b indicate that while apoE is recruited to provide CCHFV particles a ligand for LDL-R, the CCHFV GPs also allow interaction with LDL-R. These results agree with the article by Xu et al. (Cell Res, 2024) that was published during the revision of our paper and that shows an interaction between Gc and LDL-R.

Thus, based on this publication and of our new data, we removed this previous statement but we discuss in lines 376-397 the evidence that both apoE and CCHFV GPs can promote CCHFV entry via LDL-R.

e. Figure 3G, please add information to the legend as to how many technical replicates were run in this experiment? Was it a single experiment? There are no statistics included, were any of the dilutions statistically significant?

Reply: the data in this Figure were from several independent experiments. To comply with the request of this Reviewer, we repeated this experiment and plotted differently the results to show “% transduction” rather than “% inhibition”, and we revised the statistical analysis. These results are now shown in revised Figure 3f and are now statistically significant.

6. Figure 4, suggest adding negative gate (isotype control) to histogram in 4A and 4D.

Reply: we added the isotype controls to these histograms in revised Figures 5a and 5e. We also added immune-fluorescence pictures of non-permeabilized cells with vs. without either inhibitor, which shows in new Figures 5b and 5f the loss of LDL-R surface expression.

7. Figure 5, the data as presented also indicate there is an LDL-R independent mechanism of entry so this should be included in the graphical depiction.

Reply: this Reviewer is right, and we added this point in the graphical depiction (revised Figure 6).

Minor items:

1. Use of CCHFV IbAr 10200 should be corrected throughout as “wild-type” and not “full-length” virus throughout, or use CCHFV. As a suggestion CCHFV tecVLPs can also be referred to as CCHF tecVLP or CCHFVLP to avoid confusion with wild-type virus.

Reply: we changed “full-length” for “wild-type” regarding the description of the experiments with CCHFV.

CCHFV tecVLPs are now referred as CCHF tecVLPs.

2. VLPs do not infect cells as they are not replicating, they are transduced. The use of “infection” regarding VLPs leads to confusion throughout the manuscript as the authors switch back and forth regularly between the use of VLPs, pseudoviruses, and wild-type viruses. Careful editing is needed throughout the manuscript to remove the “infect” or “infection” with regards to VLPs.

Reply: we thank this Reviewer for the comments. We carefully clarified the text by using “transduction” term for CCHF tecVLPs, lentiviral vector pseudoparticles or pseudotyped VSV and “infection” for WT CCHFV and HAZV.

3. Figure 1C legend, please add the colors to the legends for CCHF tec VLP and VSVGpp.

Reply: we have modified this legend accordingly (lines 1046-1047).

4. Figure 1D, Y-axis would be more appropriately labeled as % reduction of vRNA, as it is not a direct readout of % inhibition of infection of wild-type virus.

Reply: the read-out corresponds to detection of viral RNA in infected cell lysates which reflects infection levels. We clarified this in the legend of Figure 1e (lines 1051-1054) and the Methods (lines 502-507) and Results (lines 127-132) sections.

5. Figure 1F, a more complete description of the samples is needed for the flow cytometry to make it more clear to the reader that the antibody target is on the left of the “/” and the cell type or overexpression type is on the right of the “/”. The red VDL-R/WT should state VLDL-R/WT, correct? Y axis should not state % infection, since this is VLP and the readout is the NanoLuc signal, not % infection of cells.

Reply: we clarified the legend of the flow cytometry part of the revised Figure 1f and corrected the Y-axis.

6. Line 131, Please spell out VSVGpp

Reply: we corrected the text (lines 118-119).

7. Line 146 remove typo “ca.”

Reply: we corrected the text (line 193).

8. Line 206, correct to Figure 3F.

Reply: we corrected the text.

9. Line 211, correct to 3G.

Reply: we corrected the text.

10. Add more details to the materials and methods section:

a. Include assays or references for methods of tiering wild-type or recombinant viruses used, MOIs used in experiments with infectious virus.

Reply: we clarified the relevant parts of the Method section to introduced these precisions (lines 495-582) regarding these assays with the WT (CCHFV) or recombinant (HAZV-sGFP) viruses as well as the other particles types (CCHF tecVLP, VSVpp, MLV-pp and pseudotyped VSV).

b. no methods are included for Figures 3F-H or Figure 4.

Reply: we added more details for these methods in lines 620-641, 584-591, and 737-757,

c. no methods are included for the flow cytometry of stained cells.

Reply: we added the methods lines 662-670 and 593-604.

12. Line 297-298 – As written this sentence is confusing. I assume the authors meant that the location of Lrp1 in the membrane may not be proximal to the unknown receptor so there is a possible receptor binding disadvantage because of the distance. The way it is written It sounds like the distance between the virus and cell membrane that may be suboptimal with Lrp1, does this mean there is a large size discrepancy between Lrp1 and LDL-R?

Reply: Yes, this is what we meant and we have modified this sentence to better convey this assumption (lines 407-411).

13. Line 328-330, although CCHFV is highly pathogenic it is not necessarily the most pathogenic of all the viruses in the Bunyvirales order, so this statement is unnecessary. I suggest tempering this statement to simply an important pathogenic bunyavirus or something similar.

Reply: we corrected the text accordingly (line 450).

REVIEWER COMMENTS

Reviewer #2 (Remarks to the Author):

Overall, the authors have addressed my comments and concerns, and the manuscript has been strengthened. The work remains of high interest, is timely, and warrants publication. I only have concerns about the rigor and robustness of the experiments with TyrA23, given the lack of control viruses, as noted below.

MAJOR:

Figures 2f and 2g – These experiments would have benefited from a control virus that does not bind an LDLR-related protein to strengthen the observations, particularly given how small some of the changes/phenotypes seem to be. From the response to comments by another reviewer, I can see that additional experiments were performed but that no suitable control virus could be identified, and it is suspected that the compound also has effects on other pathways in addition to LDLR (based on inhibition seen with HAZV). Rather than not including the controls in the manuscript and only showing the CCHFV data, it seems like the CCHFV data are too preliminary and would not warrant publication until a suitable control is identified. These experiments are not critical to the main conclusion of the manuscript, particularly if there is any evidence that TyrA23 acts on pathways other than LDLR. Furthermore, is the increased cell viability seen with TyrA23 treatment statistically significant? Is it possible that the higher viability makes the cells more permissive to infection?

MINOR:

1. Figure 1a can be simplified – the pictures of the instruments could be removed.
2. I would recommend the authors carefully review all axes for accuracy of what is being shown (% differences vs. fold changes). Some of the Y axes list “% of transduction” titer as the label but then axis numbers go from 0–1 or 0–2. Rather than %, is fold change being shown? (e.g., Figures 2f, 2g, and 2h). If fold change is being shown, the word “fold” should be added somewhere on the axis. Also, in Figure 1b, for quantification of blots, should the Y-axis label be % rather than fold relative to control, given that the data seem to be normalized to 100% (and the range shown is 0–120%)?

Reviewer #3 (Remarks to the Author):

The majority of the items previously described were addressed in the revised version. There are a few items that should be addressed in the revised version to improve the manuscript. Once addressed this manuscript adds to the recent developments regarding CCHFV entry and potential tropisms.

Major items:

1. Regarding the response to the description of pelleting of the VLPs through a sucrose cushion as “purifying” the virus as actin and putative cell debris was not detected in the pellets in Figure 4a – apoE is shown in the mock pellet, and actin is shown in the HAZV WB (Figure 4b). These are only two proteins that were stained for, a coomassie stained gel would likely show numerous other proteins pelleted with the virus or the mock sample.

There is also no indication that a normalization was performed for Figures 4a or 4b thus it is qualitative and quantification cannot be done, therefore the description that there was an enrichment of apoE in the CCHF tecVLP pellet is not convincing as presented and the data should not be graphed. The pulldown experiments for 4c, 4e, and 4f are much more convincing. I suggest removing 4a and 4b as they are not supportive of the claims as presented, and the data sets in 4c,

4e, and 4f are supportive.

2. For 4d, a negative control antibody is needed for IEM to ensure there was no non-specific binding of the gold-labelled secondary.

Minor items:

1. Please specify the strain used for the CCHF tec VLPs in the Materials and Methods.

2. There are a few instances where "infection" remains describing tecVLP transduction such as Lines 107, 289, 290, 1185. There may be others I have missed so please review.

3. Please remove the word "purified" with regards to sucrose pelleting of the tecVLP or HAZV: Line 273, 780, 1165, 1168

4. Figure 1h, it would be helpful to have an explanation in the text as to why there was a 24 hour incubation for the PHH cells while the timing was 48 hours for the other cell lines in Figure 1i and 1j.

5. It is unclear to me what is meant in the sentence from Line 411 to 413. Why would the rapid endocytosis of Lrp1 lead to a non-productive entry of CCHFV, if it could bind to this receptor. Other viruses use Lrp1 for entry.

6. Line 516, please define i.u. as this is the first mention of it in the manuscript.

Reviewer #2 (Remarks to the Author):

Overall, the authors have addressed my comments and concerns, and the manuscript has been strengthened. The work remains of high interest, is timely, and warrants publication. I only have concerns about the rigor and robustness of the experiments with TyrA23, given the lack of control viruses, as noted below.

MAJOR:

Figures 2f and 2g – These experiments would have benefited from a control virus that does not bind an LDLR-related protein to strengthen the observations, particularly given how small some of the changes/phenotypes seem to be. From the response to comments by another reviewer, I can see that additional experiments were performed but that no suitable control virus could be identified, and it is suspected that the compound also has effects on other pathways in addition to LDLR (based on inhibition seen with HAZV). Rather than not including the controls in the manuscript and only showing the CCHFV data, it seems like the CCHFV data are too preliminary and would not warrant publication until a suitable control is identified. These experiments are not critical to the main conclusion of the manuscript, particularly if there is any evidence that TyrA23 acts on pathways other than LDLR. Furthermore, is the increased cell viability seen with TyrA23 treatment statistically significant? Is it possible that the higher viability makes the cells more permissive to infection?

Reply: we appreciate and understand this comment. Accordingly, we have decided to remove these data from the revised manuscript.

Regarding the comment about cell viability, this Reviewer is right that we observed an increase of cell viability, which was not significant. In addition, while we observed an increase of transduction for CCHFV tecVLPs and VSVpp, we did not detect an increase of transduction for MLVpp (Figure A, below), arguing against a general increase of permissiveness of the cells.

Figure A. TyrA23 treatment.

MINOR:

1. Figure 1a can be simplified – the pictures of the instruments could be removed.

Reply: we have simplified this figure by removing the pictures of the instruments.

2. I would recommend the authors carefully review all axes for accuracy of what is being shown (% differences vs. fold changes). Some of the Y axes list “% of transduction” titer as the label but then axis numbers go from 0–1 or 0–2. Rather than %, is fold change being

shown? (e.g., Figures 2f, 2g, and 2h). If fold change is being shown, the word “fold” should be added somewhere on the axis. Also, in Figure 1b, for quantification of blots, should the Y-axis label be % rather than fold relative to control, given that the data seem to be normalized to 100% (and the range shown is 0–120%)?

Reply: we have carefully reviewed all axes for accuracy of what is being shown. Accordingly, the Y-axis label of Figure 1b is now %. Note that the graphs of Figures 2f, 2g, and 2h have been removed. The other graphs are correctly labelled.

Reviewer #3 (Remarks to the Author):

The majority of the items previously described were addressed in the revised version. There are a few items that should be addressed in the revised version to improve the manuscript. Once addressed this manuscript adds to the recent developments regarding CCHFV entry and potential tropisms.

Major items:

1. Regarding the response to the description of pelleting of the VLPs through a sucrose cushion as “purifying” the virus as actin and putative cell debris was not detected in the pellets in Figure 4a – apoE is shown in the mock pellet, and actin is shown in the HAZV WB (Figure 4b). These are only two proteins that were stained for, a coomassie stained gel would likely show numerous other proteins pelleted with the virus or the mock sample.

There is also no indication that a normalization was performed for Figures 4a or 4b thus it is qualitative and quantification cannot be done, therefore the description that there was an enrichment of apoE in the CCHF tecVLP pellet is not convincing as presented and the data should not be graphed. The pulldown experiments for 4c, 4e, and 4f are much more convincing. I suggest removing 4a and 4b as they are not supportive of the claims as presented, and the data sets in 4c, 4e, and 4f are supportive.

Reply: we agree with this Reviewer that pelleting of the VLPs through a sucrose cushion is not the best method for “purifying” virus particles, which is the reason we had also developed the other immuno-precipitation assays described in this figure.

Accordingly, we have removed the data in the panels 4a and 4b of this Figure 4.

2. For 4d, a negative control antibody is needed for IEM to ensure there was no non-specific binding of the gold-labelled secondary.

Reply: we provide in the revised Figure 4b these controls that were not shown in the previous version.

Minor items:

1. Please specify the strain used for the CCHF tec VLPs in the Materials and Methods.

Reply: the strain IbAr10200 was used to generate the CCHF tec VLPs as indicated in the Materials and Methods (lines 451 and 492).

2. There are a few instances where “infection” remains describing tecVLP transduction such as Lines 107, 289, 290, 1185. There may be others I have missed so please review.

Reply: we have changed the term “infection” in these lines, and we have reviewed throughout the manuscript that we refer to tecVLP transduction and corrected the occurrence of “infection” when inappropriate.

3. Please remove the word “purified” with regards to sucrose pelleting of the tecVLP or HAZV: Line 273, 780, 1165, 1168

Reply: the word “purified” has now been removed from these lines, as per removal of data in the panels 4a and 4b of this Figure 4. We replaced the term “purification” line 743 by “pelleting”.

4. Figure 1h, it would be helpful to have an explanation in the text as to why there was a 24

hour incubation for the PHH cells while the timing was 48 hours for the other cell lines in Figure 1i and 1j.

Reply: as compared to the other cell lines, we used a 24hour incubation for the PHH cells transduced with the CCHF tecVLP particles in order to minimize the time of culture of these primary cells (for viability reasons) and to maximize the level of nanoluc signal, as there is only one round of transduction. Of note the level of nanoluc signal in Huh-7.5 was comparable at 24h vs. 48h (see Supplementary Figure 1e and f). We have added this explanation in the text (lines 149-150).

5. It is unclear to me what is meant in the sentence from Line 411 to 413. Why would the rapid endocytosis of Lrp1 lead to a non-productive entry of CCHFV, if it could bind to this receptor. Other viruses use Lrp1 for entry.

Reply: we have clarified this sentence (line 389-391). There are examples in the literature that internalization or increased internalization to a receptor used by some viruses could be counterproductive for entry of alternative viruses. For example, the LDL-R itself, while being able to efficiently capture HCV particles, leads to a non-productive entry (Albecka et al., 2012. *Hepatology* 55:998-1007). Another example is SR-BI that also acts as a receptor for HCV: reintroducing in its C-terminal cytosolic tail a dileucine motif that is responsible for the faster endocytosis rate of SR-BII, its alternative splicing variant, reduces HCV entry and infection (Dreux et al., 2009. *PLoS Pathog* 5(2): e1000310).

6. Line 516, please define i.u. as this is the first mention of it in the manuscript.

Reply: we have changed “GFP i.u./ml” for “GFP transduction units (t.u./ml)” in the revised text and figures.